# Cyclodextrin Inclusion Complexes with Antibiotics and Antibacterial Agents as Drug-Delivery Systems—A Pharmaceutical Perspective

**DOI:** 10.3390/pharmaceutics14071389

**Published:** 2022-06-30

**Authors:** Dariusz Boczar, Katarzyna Michalska

**Affiliations:** Department of Synthetic Drugs, National Medicines Institute, Chełmska 30/34, 00-725 Warsaw, Poland; d.boczar@nil.gov.pl

**Keywords:** cyclodextrins, drugs, antibiotics, antibacterial agents, inclusion complex, host–guest complex

## Abstract

Cyclodextrins (CDs) are a family of cyclic oligosaccharides, consisting of a macrocyclic ring of glucose subunits linked by α-1,4 glycosidic bonds. The shape of CD molecules is similar to a truncated cone with a hydrophobic inner cavity and a hydrophilic surface, which allows the formation of inclusion complexes with various molecules. This review article summarises over 200 reports published by the end of 2021 that discuss the complexation of CDs with antibiotics and antibacterial agents, including beta-lactams, tetracyclines, quinolones, macrolides, aminoglycosides, glycopeptides, polypeptides, nitroimidazoles, and oxazolidinones. The review focuses on drug-delivery applications such as improving solubility, modifying the drug-release profile, slowing down the degradation of the drug, improving biological membrane permeability, and enhancing antimicrobial activity. In addition to simple drug/CD combinations, ternary systems with additional auxiliary substances have been described, as well as more sophisticated drug-delivery systems including nanosponges, nanofibres, nanoparticles, microparticles, liposomes, hydrogels, and macromolecules. Depending on the desired properties of the drug product, an accelerated or prolonged dissolution profile can be achieved when combining CD with antibiotics or antimicrobial agents.

## 1. Introduction

Due to the high costs associated with the development of innovative drugs, a very long time to introduce them to the market, and a huge risk of failure in clinical trials, more and more work is focused on the development of new drug-delivery systems [1] based on existing active pharmaceutical ingredients (APIs) with a known safety profile and physicochemical properties. Currently, one of the directions of intensive research in the field of designing new forms of drugs is the development of new delivery systems using auxiliary substances to modify the unfavourable physicochemical properties of known APIs, and thus obtain a better therapeutic effect. In this respect, cyclodextrins (CDs) have undoubtedly been successful.

CDs are a valuable subject of research in many areas, including pharmacy [2], food chemistry, cosmetics, solar energy utilisation, and environmental applications [3,4], as well as separation of enantiomers in analytical chemistry [5]. The global technological landscape of CDs for pharmaceutical applications and the evolution of those technologies over time was presented by Rincón-López et al., using patents as the technical source [6]. Currently, there are over 50 CD-based formulations on the market for a variety of therapeutic purposes, including four containing antibiotics and antibacterial agents such as cefotiam, cefditoren pivoxil, chloramphenicol, and norfloxacin [7]. To date, there are many excellent review articles available on native CDs and their derivatives, their well-known structural features, physicochemical properties, and their possible applications [8,9]. We will therefore characterise these topics in a very limited scope and direct readers to numerous reviews in this field.

Native CDs consist of 6, 7, or 8 glucopyranose units, referred to as α, β, and γ, forming a macrocyclic ring. One of their most important properties is that their shape is similar to a truncated cone, with a hydrophobic inner cavity and hydrophilic surface. This enables them to form inclusion complexes (ICs) with different molecules using non-covalent interactions, such as the host–guest interaction and hydrogen bonding. Many derivatives of CDs have been synthesised but only three (hydroxypropyl, sulfobuthyl ether, and randomly methylated β-CD derivatives—HP-β-CD, SBE-β-CD, and RM-β-CD, respectively) are frequently used as excipients due to their proven non-toxicity [10]. Depending on the type of CD, there are different allowable routes of administration of the complex, such as oral, nasal, rectal, dermal, ocular, or parenteral [10]. Hence, the complexes can be prepared in a solid form, in solution, or by grafting the CDs onto various surfaces. In this article, we will look at combinations of CDs with antibiotics or antibacterial agents, wishing to emphasise that such combinations play a key role in improving already approved drugs. This trend is expected to continue in the coming years due to the versatility of drug-delivery systems and the fascinating ability to improve the properties of APIs as well as to develop pharmaceutical formulations in the design of innovative therapies. We expect that such an approach will be desirable in the context of antibiotics, as approved antibiotics gradually lose their activity as a result of an increase in the number of drug-resistant bacteria, and simultaneously a small number of new drugs enter clinical practice. To the best of our knowledge, no comprehensive summary of antibacterial drug complexation with CDs has been published so far and only selected examples of antibacterial drug/CD complexes are provided in the available review articles [11,12]. Reports of CD complexes of antibiotics from nearly all classes are discussed in this article, ranging from pioneering works to significant articles published online by the end of 2021.

Our review focuses on drug-delivery applications such as improving solubility, modifying the drug-release profile, slowing down the degradation of the active substance, improving biological membrane permeability, enhancing antimicrobial activity, and masking the bitter taste. Works that only deal with the structures of the complexes are also included, as knowledge of the structure can aid in understanding macroscopic properties at the molecular level. Some antibiotic molecules are too large to be included in the CD cavity; however, due to the improvement of pharmaceutical properties, they are also presented in this article. On the other hand, our study omits reports in which CDs were used as constituents of a pharmaceutical formulation but their influence on the system’s properties was not investigated.

### 1.1. The Use of CDs and Their Derivatives as Drug Carriers

The particular interest in CDs in the field of pharmaceutical applications results from their ability to modulate several properties that influence the action and therapeutic profiles of drugs. The main purpose of utilising CDs and their derivatives as drug carriers is to increase the solubility of drugs belonging to Class 2 (low solubility, high permeability) or Class 4 (low solubility, low permeability) of the Biopharmaceuticals Classification System (BCS) [13], ultimately increasing the bioavailability of poorly water-soluble drugs by improving their release profile [14]. Interestingly, Carrier et al. attempted to answer the question regarding the properties of compounds whose bioavailability is increased using CDs [15]. The authors conducted a critical review of the literature, looking for a correlation between the physical and chemical properties of a drug and the possibility of increasing bioavailability using CDs, mainly focusing on the solubilising capabilities of CDs for small molecular weight compounds. The obvious requirement for increasing bioavailability is the formation of a complex between the drug and CDs. The parameters that were critically assessed during the review, which may or may not lead to an increase in the bioavailability of the drug when complexed with CD, were: hydrophobicity, solubility, drug charge during gastrointestinal transit, and factors such as the magnitude of binding constant, dissolution kinetics of the compound, drug dose, interaction with other components, method of complex formation, type of CD used, and the resulting molar ratio. The authors provided general guidelines but postulated that a more comprehensive theoretical analysis of the effect of CDs on kinetics was needed to distinguish between test parameters that showed success or those that showed no enhancement in the bioavailability.

In some cases, the complexation of a drug with a CD can also increase the permeability through biological membranes and the effects of the type and concentration of the CD on the interactions with different types of cell membranes and biological barriers have been extensively described [16]. Given the hydrophilic nature, high molecular weight, low octanol/water partition coefficient of CD, as well as the number of hydrogen bond donors and acceptors, CD-drug complexes mainly enter cells by endocytosis (e.g., in the form of nanoparticles). Therefore, the development of new CD-based carriers targeting the endocytic pathway could help improve drug penetration through biological barriers [16]. However, according to in silico simulations [17], β-CD and its assemblies can often improve the water solubility of lipophilic drugs, enhancing the bioavailability of the guest molecule by transporting it directly to the barrier membrane, where the complex dissociates and the free drug can be absorbed [18]. This is the case when the binding forces between the CD and the complexed drug are not very strong, otherwise complex dissociation cannot take place. The tests assessing the ability of medicinal substances to penetrate through biological membranes use both the non-cell-based parallel artificial membrane permeability assay (PAMPA) and cell-based (e.g., Caco-2 and mucus-producing Caco-2 co-culture cell model) permeability models. There are several possible mechanisms of CD-mediated drug permeability, depending on the hydrophobicity of CD. Lipophilic CDs such as the methylated derivatives are thought to increase drug flux by altering the barrier properties of the membrane through component extraction or fluidisation. The hydrophilic CDs also modulate drug flux through membranes, but via different mechanisms. The first relates to increasing the total amount of dissolved API molecules in the aqueous donor phase, thereby increasing the concentration gradient and the availability of free molecules on the membrane surface, leading to faster penetration of these molecules into the acceptor phase (lipophilic membrane surface). Second, by preventing the interactions between API and glycoproteins such as mucin (rheologically important water-soluble biopolymer), CDs increase the overall delivery rate of the drug substance to the membrane surface. Finally, CDs behaving like chaotropes at relatively low concentrations disrupt the network of hydrogen bonds between water molecules, which may facilitate the availability of CD/API complexes on the membrane surface [19].

### 1.2. New Possibilities of Using CDs

Formulations based solely on CDs do not have any intrinsic mechanism to control the release of the drug (most such systems are immediate-release dosage forms); therefore, smart nanosystems have been created in combination with various nanomaterials to control the drug release. These systems are very frequently designed to release at a specific site, through the use of internal (pH, reductant environments, enzymes) and external (light, magnetic field, ultrasound) stimuli, which have been discussed in detail in the reviews of Haimhoffer et al. [16] and Real et al. [20].

The release of API from the polymeric delivery systems can be governed by diffusion or the processes of swelling or relaxation of the polymeric chain [21]. To determine which of them is slower and therefore determines the rate of the drug release, the curve obtained from one of the possible kinetic models can be fitted to the experimentally obtained dissolution profile. The so-called power laws, also referred to as Ritger–Peppas or Korsmeyer–Peppas models, can be described by the general formula:(1)$$\frac{M_{t}}{M_{\infty }}=Kt^{n}$$
where $M_{t}$ is the amount of drug released at time t, $M_{\infty }$ is the amount of the drug at the equilibrium, K is the constant which incorporates the structural modifications and geometrical characteristics of the system, and n is the exponent. Depending on the value of n that better adjusts to the release profile, the classification can be established. For n = 0.5, the drug release is governed by diffusion (Fickian model), and Equation (1) is called a Higuchi model. When n = 1, the drug-release rate corresponds to zero-order kinetics and the mechanism driving the process is the swelling or relaxation of polymer chains (a so-called Case II transport). When 0.5 < n < 1, the model is non-Fickian or anomalous transport, and then both diffusion and swelling determine the overall rate [21].

Moreover, despite the many advantages of CDs, there is one disadvantage that limits their use in solid dosage forms, which relates to large amounts of CDs that are required to solubilise a drug, thus increasing the formulation bulk. To overcome this obstacle, multicomponent complexes have been developed.

Ternary complexes with the addition of auxiliary substances such as (i) amino acids, (ii) organic acids and bases, and (iii) water-soluble polymers have been extensively explored to increase complexation efficiency, and to stabilise or prevent aggregation of the complexes [22]. The water-soluble polymers include polyvinylpyrrolidone (PVP), hydroxypropylmethylcellulose (HPMC), poloxamers, chitosan, hyaluronic acid, polyethylene glycol (PEG), Aerosil^®®®^, and carboxymethyl cellulose (CMC). Other multicomponent complexes with organic acids such as citric, gluconic, tartaric, lactic, and malic acids, or with organic bases such as three mono-, di-, and tri-ethanolamines or amino acids (glycine, cysteine, proline, arginine, lysine, aspartic acid, and glutamic acid), have already been described [22]. The use of amino acids in combination with CD and an antibiotic appears to be of particular interest, due to the documented increase in antimicrobial activity and the decrease in the toxicity of the drugs [23].

Another modern approach in the use of CDs is the development of CD-modified nanomaterials for drug delivery. Systems consisting of CDs, drugs, and lipid-based polymerics or surface-modified nanocarriers have been described by Real et al. [20]. Recently, an excellent overview of nanomaterials for antimicrobial applications in terms of types, mechanism of action, antibiotic therapeutic uses, and toxicity considerations has been discussed by Ndayishimiye et al. [24]. The authors note that, in the case of nanoparticles (NPs), bacteria may not be able to rapidly develop resistance to NPs because the necessary resistance mechanisms are no longer found in the natural offensive/defensive arsenal of bacteria. Interestingly, NPs, due to their propensity to enter the bilayers of the bacterial cell membrane, allow them to reach the cytoplasm while disrupting membrane function and integrity, thus acting as inherent antimicrobial agents. The readers interested in this topic are referred to [24]. Nanoparticles generally vary in size from 10 to 1000 nm, which makes it easier for them to pass through cell barriers to reach their target. In drug-delivery systems, NPs should be mucoadhesive, biocompatible, non-toxic, and biodegradable. They can be synthesised by various techniques such as ion gelation, emulsification-cross-linking, coacervation, and solvent evaporation.

Hydrogels are another approach [25]. The use of hydrogels as components of medical devices and drug-device combination products is becoming more and more popular. Hydrogels are mainly made of biopolymers and/or polyelectrolytes. They can be divided into two types: those made of natural polymers (agar, carrageenan, sodium alginate, collagen, chitosan, cellulose, starch, and gums) and those made of synthetic polymers, such as polycaprolactone (PCL), poly(vinyl alcohol) (PVA), PEG, PVP, and poly(lactic acid). Hydrogels synthesised from natural polymers are delicate, hence the natural polymers are cross-linked, grafted with monomers, or blended with synthetic polymers to improve their mechanical properties. Another classification of hydrogels depends on the ionic charges on the groups bound, introducing cationic, anionic, or neutral hydrogels. The classification criteria can also relate to the types of cross-linking method or their response to stimuli. Hydrogels can be formed through physical, chemical, and hybrid bonding. Due to non-covalent forces, physical gels can change from liquid to gel in response to changing environmental conditions such as temperature, ion concentration, pH, or other conditions. On the other hand, a covalent bond between the polymer chains is used in chemical gels, introducing mechanical integrity and resistance to degradation compared to other weak materials. Chemical cross-linking occurs by various means, utilising small molecules (formaldehyde, glutaraldehyde, genipin, diglycidyl ether, and *N*,*N*′-methylenebisacrylamide). The basic concepts of hydrogels, in terms of their synthesis, properties, and applications, have been extensively described in an excellent review by Bashir et al. [25]. Nevertheless, the inherently high water content, necessary for biocompatibility, limits the success of hydrogels as drug-delivery systems. Only polar drugs can be efficiently introduced into the water phase of hydrogels, and when administered to the body, release is usually too fast for therapeutic purposes. For this reason, the formation of a CD IC with various drugs gives hydrogels a unique mechanism to control drug loading and delivery. To do this, CDs have to be fixed to the network cross-linked by condensation [26].

### 1.3. Expectations for the Drug-Delivery Systems Containing Antibiotics and Antibacterial Agents

Antibiotics are very diverse in terms of their antibacterial activity, physicochemical properties, polarity, target point, side effects, and chemical stability. For example, fluoroquinolones and tetracyclines suffer from photodegradation, whereas beta-lactams and tetracyclines are unstable in aqueous solution. It is therefore particularly important to stabilise such sensitive drugs, especially when administered parenterally [11]. There are many reported examples showing how CDs can inhibit the degradation of different drugs [27]; however, some reports also highlight situations in which the presence of CDs led to the acceleration of antibiotic degradation [28,29].

Antimicrobial activity is expressed as a function of time or concentration [30]. Antibiotics such as beta-lactams, macrolides, and glycopeptides are time-dependent, which means that the duration of a sufficiently high concentration is crucial for their efficacy, while aminoglycosides and fluoroquinolones are concentration-dependent drugs, meaning that the maximum concentration is the most important, and their duration has a limited impact on antibacterial efficacy. For this reason, the combination of CDs with antibiotics or antibacterial agents meets different expectations, namely, an accelerated or prolonged dissolution profile, depending on the desired properties of the drug product [31].

Due to the growing drug resistance among bacteria, it is very important to increase antimicrobial efficacy, which can occur through the spontaneous formation of hydrogen bonds between CDs and several components of the bacterial cell membrane [32]. Hence, complexation with CDs enables antibiotics to be used in much lower concentrations, thus reducing exposure to the antibiotic, which in turn delays the onset of antibiotic resistance [12,33,34]. A recent study correlated antimicrobial activity with the hydrophobicity/hydrophilicity balance of polyalkylamino CDs [35]. Relatively hydrophilic CDs (log *p* < −0.1) may interact with the bacterial cell membrane’s anionic surface but cannot penetrate into the lipid bilayer membrane’s apolar hydrocarbon core, while relatively hydrophobic CDs (log *p* > 0.1) may disturb the surface interaction. The authors concluded that only balanced derivatives can disrupt the bacterial membrane [35]. The study suggested that the differences in the cell membrane and lipid molecules of Gram-positive and Gram-negative bacteria may contribute to the varied antimicrobial activities of CDs.

## 2. Beta-Lactam Antibiotics

Beta-lactam antibiotics include numerous compounds characterised by the presence of a beta-lactam ring in the molecule, which is responsible for the antibacterial activity. This group includes penicillins, cephalosporins, carbapenems, and monobactams, as well as beta-lactamase inhibitors [36]. The physiochemical properties of such a chemically diverse set of APIs can be discerned by the route of administration, taking into account the distinct solubility and permeability requirements of intravenous and oral drugs [37]. Antibiotics of this group are rather polar and hydrophilic [37], and hence they hardly cross biological barriers, including the blood–brain barrier. As mentioned in the Introduction Section, the action of beta-lactam antibiotics is time-dependent, and thus the measure of efficacy is the time period in which the concentration of the drug remains above the minimum inhibitory concentration (MIC) [30]. Hence, there is a need to administer beta-lactam antibiotics several times a day, and the tendency to reduce the number of doses per day therefore emerges, which is usually justified by a better patient compliance. Thus, complexes of beta-lactam antibiotics with CDs are desirable to allow the safe use of the sustained-release pharmaceutical form in a single daily dose.

### 2.1. Penicillins

Penicillins, historically the first antibiotics to be discovered, contain a so-called penam moiety, i.e., fused beta-lactam and thiazolidone rings (Figure 1). Some penicillins can be obtained by biosynthesis, and a much more numerous group is compounds obtained by semi-synthesis from 6-aminopenicillanic acid (6-APA), which is the starting compound for the preparation of further derivatives. Penicillin obtained by biosynthesis is sensitive to digestive juices, which makes it impossible to administer orally, and it is also characterised by a narrow range of action and sensitivity to beta-lactamases. Only some penicillins are suitable for oral administration, but (with the exception of amoxicillin), their absorption is not perfect. The malabsorption of most penicillins is partly related to their low lipophilicity. Penicillin is not very stable in solution, resulting in the formation of not fully defined degradation products, and this could be responsible for the occurrence of allergic reactions, which are often very violent.

Detailed information on the types of CDs used, the methods of complex preparation, and the techniques used to characterise the obtained products in each indicated article are summarised in Table 1. Both binary, ternary, and new drug-delivery systems are listed in the table to provide a comprehensive view of the individual compounds.

**Benzylpenicillin (BP)** (known as **penicillin G**) undergoes hydrolysis of the beta-lactam ring in aqueous solution and has a short biological half-life as well as low oral bioavailability, which makes its use difficult. Therefore, the main goal of complexation with CDs is to prevent its degradation. Pop et al. [38] improved the stability of BP by complexing with HP-β-CD to create a delivery system that is stable to oxidation, acid-catalysed hydration, and hydrolysis, which occurs mainly at alkaline pH. Kinetic studies were continued by Ong et al. [39], who investigated the influence of the drug:HP-β-CD ratio on the stability of penicillin G and determined thermodynamic parameters, such as activation energy, enthalpy, and entropy, while Hada et al. [40] extended the scope of the study to α-, β-, and γ-CD. The degradation of BP complexed with HP-β-CD was about nine times slower than that of un-complexed BP [39]. Aki et al. [41] investigated the structure of the complexes of BP with HP-β-CD using microcalorimetry, nuclear magnetic resonance (NMR) spectroscopy, and molecular modelling. Two different types of IC with a 1:1 stoichiometry ratio were formed in a strong acid solution: Complex I with the phenyl ring of BP penetrated into the HP-β-CD cavity, and Complex II with the penam moiety included in the cavity by hydrophobic interaction.

Popielec et al. [42] investigated the effect of 13 different neutral, positively, and negatively charged CDs, finding that HP-β-CD and RM-β-CD had a stabilising effect on the beta-lactam ring in aqueous acidic solution but generally accelerated the hydrolytic decomposition of this ring in neutral and basic conditions. Another study investigated the effect of methylation of CD on the chemical stability of a complex with BP prepared in the solid state using the freeze-drying method [43], demonstrating that fully methylated β-CD stabilised BP, while the partial methylation of β-CD only reduced the catalytic effect of native β-CD to some extent.

**Phenoxymethylpenicillin** (PMP) (also known as **penicillin V**) was combined with β-CD by a saturated aqueous solution method to mask the unpleasant smell of this pharmaceutical raw material [45].

**Ampicillin** is one of the best-studied beta-lactam antibiotics with regard to the formation of CD complexes, due to its low bioavailability. Like BP, ampicillin is hydrolysed in aqueous solution to form polymers with strong antigenic properties. Meanwhile, many bacterial strains produce beta-lactamases, i.e., enzymes that hydrolyse the beta-lactam ring, causing its inactivation and the formation of degradation products. Accordingly, the formation of an IC with CD can directly limit the development of bacterial drug resistance by inhibiting the degradation of ampicillin. The inhibitory effect of β-CD on ampicillin polymerisation in aqueous solutions was found by fast-atom bombardment mass spectrometry (MS) and circular dichroism spectroscopy [46], whereas spectrophotometric and vapour pressure osmometric methods were used to demonstrate that β-CD improved the solubility, dissolution rate, and bioavailability of the drug [47]. Aki et al. [48] investigated the inhibitory effect of pH and the CD type (β-CD, HP-β-CD) on the stability of ampicillin and analysed the structures of the resultant complexes. In a strong acid solution, cationic ampicillin and CDs formed two different types of ICs with a 1:1 stoichiometry: one including the penam moiety of ampicillin in the CD cavity (Mode I), and the other including the phenyl group. The zwitterion and anion of ampicillin only formed one type of IC with CD, which included the phenyl and the penam group, respectively. Maffeo et al. [34] conducted an NMR and MS study of the interaction between beta-lactam antibiotics, including ampicillin and dicloxacillin, and neutral (β-CD, γ-CD) or carboxylated, negatively charged CDs, to better define host–guest interactions and evaluate the effect of the CD on the rate of hydrolysis. Upadhyay and Ali [49] conducted 2D COSY (COrrelation SpectroscopY) and ROESY (Rotation Frame Nuclear Overhauser Effect SpectroscopY) NMR studies, which confirmed that the phenyl moiety of ampicillin is included inside the hydrophobic cavity of the β-CD, and determined the mode and depth of inclusion. Namazi and Kanani [50] synthesised a telechelic polymer of β-CD-poly(ethylene glycol) to attach three beta-lactam antibiotics, including ampicillin, amoxicillin, and cefalexin, thereby forming new prodrugs. The dissolution profiles indicated that the isolated prodrugs could have potential applications for controlled drug release as oral drug-delivery systems. Lampropolou et al. [51] suggested a new approach, namely the inclusion of ampicillin into β- and γ-CD connected on their narrow side with monosaccharides such as D-mannose and D-*N*-acetylglucosamine. These monosaccharides are recognised by bacterial lectins, i.e., carbohydrate-binding proteins which are highly specific for sugar groups, and the resulting complexes therefore interacted more strongly with lectins immobilised on the surface.

**Pivampicillin**, a prodrug of ampicillin, was one of the three test beta-lactam antibiotics (in addition to cefuroxime axetil and cefetamet pivoxil) selected by Mizera et al. [53] to develop a new machine learning algorithm. Based on the infrared spectra of the test sample and the pure ingredients, the model would predict whether an IC (with α-CD, β-CD, γ-CD, or their hydroxypropyl derivatives) or a physical mixture was formed. The analysis of the contribution of spectral bands to the model showed the interactions of the ester group of the beta-lactam antibiotics and the penam moiety of pivampicillin with the CDs. The machine learning model adequately separated the samples of pivampicllin that formed the IC from those that did not.

**Amoxicillin** complexes with CDs have been studied by many research groups, mainly to propose a delivery system that will help stabilise amoxicillin in an acidic environment and support the more effective use of amoxicillin to eradicate *Helicobacter pylori* in the treatment of gastric and duodenal ulcers. NMR spectroscopy confirmed the complexation of amoxicillin with β-CD and γ-CD, while α-CD only had a very small effect on the chemical shifts of amoxicillin [54]. Aki et al. [56] investigated the inhibition of the acidic degradation of amoxicillin induced by HP-β-CD. Since amoxicillin, like ampicillin, exists as a cation, zwitterion, or anion, the formation of an IC in aqueous solution is influenced by the pH of the solution. In strong acid solution, cationic amoxicillin and CD formed two different types of ICs with 1:1 stoichiometry, similar to penicillin G: one containing the penam moiety of amoxicillin in the CD cavity (Mode I), and the other containing a phenyl group. A complex with a stoichiometry of 1:2 (amoxicillin:CD) was also formed. Since the beta-lactam ring of amoxicillin was protected by the inclusion into HP-β-CD, drug degradation was approximately four times slower than for amoxicillin alone at acidic pH, effectively increasing the stability of amoxicillin from the acidic degradation. In another study, isothermal titration microcalorimetry and a petri dish bioassay method were used to demonstrate that the antibacterial activity of the amoxicillin complexes can be arranged in the following order: amoxicillin/β-CD > amoxicillin/γ-CD ≈ amoxicillin/α-CD > amoxicillin [57]. As β-CD protects the penam ring from degradation, the antibacterial activity of amoxicillin/β-CD complex was mainly related to the increased chemical stability of the antibiotic in the strongly acidic solution.

Solubilisation and stabilisation of **dicloxacillin** complexes with α-, β-, γ-, and HP-β-CD were also studied at different pH levels [59]. The IC with γ-CD had the highest stability constants at pH 1 and 2, while HP-β-CD at pH 3. To comprehensively understand the effects of the inclusion of **cloxacillin** sodium into β-CD, the complex was examined in solution by NMR and fluorescence and then obtained in the solid state by the co-precipitation method [60]. The results confirmed the existence of 1:1 stoichiometry and the inclusion of a chlorophenyl group into the inner cavity of the β-CD. **Methicillin** in combination with a specially designed β-CD derivative, namely per-6-(4-methoxylbenzyl)-amino-6-deoxy-β-CD HCl salt, showed an increased affinity for a resistant target such as altered penicillin-binding protein (PBP) 2a, which resulted in the recovery and enhancement of its antibacterial activity against methicillin-resistant strains of *Staphylococcus*
*aureus* (MRSA) [61]. Since the efficacy of beta-lactam antibiotics relies on their ability to reach and bind the PBPs, MIC values of 2.0~4.0 mg/L were obtained against two MRSA strains for the methicillin/per-6-(4-methoxylbenzyl)-amino-6-deoxy-β-CD, compared to more than 128 mg/L for methicillin alone and more than 64 mg/L for the commercial methicillin/HP-β-CD combination. **Oxacillin**, which is anionic at physiological pH, was combined with specially designed cationic CD derivatives [62]. Isothermal titration calorimetry (ITC), NMR, and in vitro tests were performed, with the positively charged octakis-(6-(2-aminoethylthio)-6-deoxy)-γ-CD forming the most stable IC with oxacillin, which can be considered as a promising vehicle for the protection and delivery of this antibiotic.

### 2.2. Cephalosporins

Cephalosporins constitute a subgroup of beta-lactam antibiotics in which the beta-lactam ring is fused with a dihydrothiazine ring, thereby forming a so-called cephem moiety (Figure 1). This large class of drugs is divided into several generations, depending on the attached substituents and the spectrum of activity against Gram-positive and Gram-negative bacteria. A summary of the experimental details of the combinations with CDs is presented in Table 2.

To understand the basic aspects of the **cefalexin** (the first generation of cephalosporins) binding mechanism to the β-CD cavity, structural characterisation was performed by NMR, molecular mechanics, quantum mechanical calculations, and molecular docking [64].

**Cefuroxime axetil** (a representative of the second generation of cephalosporin) is a prodrug of cefuroxime that belongs to BCS Class II with a bioavailability of 37% after oral administration on an empty stomach [95]. Cefuroxime axetil was complexed with β-CD by the kneading method, leading to 9.3-fold increase in the dissolution rate compared to the parent drug [67]. Mizera et al. [53] obtained cefuroxime axetil ICs with α-, β-, γ-, HP-β-, HP-γ-, and methyl-β-CD (M-β-CD) by the co-precipitation method, while only a physical mixture was obtained in the case of HP-α-CD. Spectral analysis showed that this antibiotic bound mostly with the axetil group, where interactions of carbonyl groups in the (acetyloxy)ethyl group and ((aminocarbonyl)oxy)methyl group were observed. A machine learning algorithm was developed, relying on infrared spectra of the pure ingredients and their combinations, to determine whether a complex or a physical mixture was formed. However, the system predicted a false-negative result for the cefuroxime axetil/HP-β-CD complex, which may suggest that the system under consideration has a unique binding mode across the dataset, which cannot be predicted based on other examples. In a subsequent report, a theoretical model based on docking and molecular modelling was developed, which predicted that HP-β-CD would form the most thermodynamically preferred system with cefuroxime axetil [68]. Indeed, the complexation with HP-β-CD was shown to significantly improve dissolution profiles compared to the parent cefuroxime axetil and with antimicrobial efficacy increased up to 4-fold.

Another study showed the affinity order of four CDs for cefuroxime axetils as follows: HP-β-CD ~ γ-CD > β-CD ~ α-CD [69]. Raman spectroscopy and mapping, along with a molecular dynamics simulation, showed that α- and β-CD interact with the furanyl and methoxy moieties of cefuroxime axetil, and γ-CD creates a more diverse interaction pattern with the parent drug, while HP-β-CD binds cefuroxime axetil with the contribution of hydrogen bonding. Shah et al. [70] compared binary complexes of HP-β-CD/cefuroxime axetil with ternary ones, in the presence of certain auxiliary substances such as PVP K-30, HPMC, poloxamer 188, and/or PEG 4000, along with Aerosil^®®®^ 200. The ternary systems performed better than the binary systems due to the synergistic effect of ternary complexation and the particle size reduction achieved by the spray-drying technology. The stability constant for the combination of HP-β-CD/cefuroxime axetil with or without the addition of polymer can be set in the following order: PEG 4000 > poloxamer 188 > HPMC > PVP K-30 > binary system. A similar attempt to include cefuroxime axetil in β-CD with and without L-arginine showed that the ternary system performed better than the binary system, offering much better solubility and dissolution rate than cefuroxime axetil alone [71]. The ternary system was also investigated as a combination of cefuroxime axetil, β-CD, and PVP K-30, and showed the best properties compared to the binary complex and the pure drug substance, with in vitro release tests >85% dissolution within 30 min [72].

Complexes of **cefaclor** with β-CD were prepared and their stability was compared to a parent drug by means of the thin-layer chromatographic (TLC) densitometric method, using such parameters as the rate constant and activation energy [73]. β-CD showed the strongest effect on stability of cefaclor in 0.5 mol/L of HCl, while both higher and lower acid concentrations decreased the influence of β-CD on the amount of cefaclor in the examined samples.

**Ceftazidime** (a representative of the third generation of cephalosporin) was included into HP-γ-CD by the freeze-drying method [74]. In another study, β-CD or HP-β-CD complexes of ceftazidime were prepared by solvent evaporation, the kneading method, or a physical mixture to prevent hydrolytic degradation upon reconstitution with water [75]. Better results were obtained with HP-β-CD complexes prepared by the kneading method, which were then used to create a stable ceftazidime formulation with poloxamer 127 and cremophore RH, allowing for a lower hydrolysis rate constant, improved shelf life (about 33 days in pH 5.5 and pH 7.4), and good antimicrobial activity.

**Cefixime** (BCS class IV) was combined with Captisol^®®®^, which is a patent-protected variation of SBE-β-CD whose chemical structure is designed to enable the creation of new products by significantly improving the solubility, stability, and bioavailability. The addition of hypromellose had a stabilising effect on the cefixime-Captisol^®®®^ complexation, reducing the degradation rate by approximately 1.5 times (in the range of 303–353 K), while povidone K-30 and macrogol (PEG) 4000 had an unfavourable effect due to steric hindrance, preventing the guest from entering the CD cavity [78]. Pamudji et al. [79] applied high-performance liquid chromatography (HPLC) to perform a stability study at 40 °C and 75% RH in a liquid suspension. The authors showed that the best cefixime:β-CD ratio was 1:2, and the kneading method proved to be more suitable than freeze-drying, as the latter decreased the stability of cefixime. A further study compared two different approaches to increase the solubility of cefixime, namely a solid dispersion using Pluronic, which is an inhibitor of P-glycoprotein, important in inhibiting intestinal drug efflux, and β-CD inclusion complexation [80]. Both were found to increase solubility, but the use of Pluronic with the spray-drying method was best-suited to achieve enhanced dissolution. However, drug permeation using the rat non-everted intestinal sac method (ex vivo) showed that Pluronic was not involved in permeability enhancement of cefixime. The magnitude of the difference between the two approaches (Pluronic—solid dispersion and β-CD—IC) was not due to the ability of the Pluronic to enhance permeation, but only to its ability to improve solubility [80]. Another study found that the spray-drying method achieved the greatest solubility improvement of the cefixime complex with HP-β-CD compared to other methods, including solvent evaporation, co-grinding, microwave irradiation, and freeze-drying [81]. These studies showed that the binary system of cefixime showed greater antimicrobial activity against *S. aureus* and *E. coli* than cefixime alone.

A different approach to improve the solubility and dissolution rate of cefixime was the preparation of binary and ternary systems by the freeze-drying method [82]. The results confirmed that the combined use of β-CD and hydrophilic polymers (PVP K-30 at 0.25% *w*/*v*, HPMC K4 at 0.1% *w*/*v*) significantly improved the drug solubility and dissolution rate, due to the increased efficiency of CD complexation. The increase in the dissolution rate was found to be higher for the ternary systems than for the corresponding binary compositions, and the ternary systems with HPMC were found to exhibit a faster dissolution profile compared to other systems. The apparent stability constants of cefixime with binary and ternary systems were ranked as follows: cefixime-β-CD-HPMC > cefixime-β-CD-PVP > cefixime-β-CD. Similarly, a synergistic effect of L-arginine in combination with cefixime and β-CD or HP-β-CD resulted in the ternary system being most effective, resulting in a tremendous improvement in physicochemical properties (which was better for the HP-β-CD) compared to the pure drug [83]. In addition, molecular modelling studies confirmed the stoichiometry, stability, and geometry of the cefixime/β-CD complex with the inclusion of the cephem moiety inside the cavity from the wide rim in the binary complex, and the thiazolyl ring embedded in the CD cavity through the narrow rim in the ternary system. Recently, Cirri et al. [84] investigated the effect of 3 different CDs (γ-CD, HP-β-CD, and SBE-β-CD) and 11 amino acids on the solubility and taste of cefixime formulations. Solubility studies showed that SBE-β-CD and histidine were the most effective combination for an oral cefixime solution for use in children, although no synergistic effect was found that would improve the solubility of cefixime when these excipients were used together. The solubility of cefixime increased from 0.76 to 17.5 mg/mL. Antimicrobial activity was almost unchanged, with a slight increase of MIC values over time. NMR spectroscopy was used to gain insight into the interactions between components in the binary or ternary systems.

**Cefetamet pivoxil** was analysed as one of three model beta-lactam antibiotics to develop a machine learning algorithm [53]. The most likely binding mode involved interactions of the pivoxil group and some interaction between the thiazole ring and the cephem group with polar, hydroxypropyl-rich moieties located on the outer side of the CD. The algorithm predicted one false-negative and one false-positive result for systems containing cefetamet pivoxil.

**Cefpodoxime proxetil** (CPP) (BCS Class II/IV) was complexed with β-CD in a 1:2 molar ratio. The freeze-dried product exhibited the highest dissolution rate at almost 90% after 30 min compared to 34% and 42% for the physical mixture and solvent evaporation, respectively [85]. In vitro studies against *Neisseria*
*gonorrhoeae* strains showed an enhanced therapeutic efficacy of the prepared freeze-dried complexes. Recently, a spray-drying technique was proposed instead of kneading to obtain CPP complexes with β-CD, HP-β-CD, and M-β-CD with a better release profile and enhanced antibacterial activity [86]. The solubility of ICs increased from 3 to 5 times. The interactions for CPP, arranged in order from strongest to weakest, are as follows: HP-β-CD > M-β-CD > β-CD. The MIC values for the *E. coli* and *Klebsiella pneumoniae* of binary systems were found to be two to four times lower than that of CPP. Meanwhile, the MIC values for *S. aureus* of the complexes with HP-β-CD and M-β-CD were two times lower than that of pure API, while the complex with β-CD displayed the same MIC value as the pure CPP.

Another study proposed the complexation of CPP with HP-β-CD with the addition of a soluble polymer, sodium CMC, to form a ternary system [87,88]. Not only was the solubility of the drug increased, but also the permeability through biological membranes (study on Caco-2 cells’ monolayers). The permeability studies showed that the complex of CPP with HP-β-CD in the presence of polymer has a higher permeability compared to CPP alone and the complex of CPP with HP-β-CD in the absence of a polymer [87]. The complex tablets were prepared by direct compression, showing the increased solubility, dissolution rate, and antimicrobial activity compared to commercial tablets [88].

**Cefdinir** (oral bioavailability of 21–25%) was kneaded with β-CD and HP-β-CD in a 1:1 molar ratio, showing that β-CD had a greater effect on solubility enhancement (101% for β-CD versus 23% for HP-β-CD), while HP-β-CD allowed for better antimicrobial activity [89]. In other studies, the kneading method was used alongside other methods, namely co-evaporation, spray-drying, and microwave irradiation, to obtain β-CD [90] and HP-β-CD [91] complexes in the solid state. It was demonstrated that spray-drying and microwave irradiation resulted in the formation of a true complex, with the complexes prepared by the microwave irradiation method being obtained with the best yield and the highest dissolution rate. Another article reported on a complex with HP-β-CD prepared by the freeze-drying technique, which improved the solubility of cefdinir 2.36 times [92]. Recently, Morina et al. [93] showed increasing solubility of cefdinir/CD complexes, as indicated by phase solubility studies (PSS), in the following order: β-CD < HP-β-CD < γ-CD < SBE_7_-β-CD. The MIC results against *S. aureus* and *E. coli* showed that the formation of ICs, especially with HP-β-CD, enhanced the antimicrobial activity of the drug, which is consistent with the results [89]. Subsequently, an oral tablet with cefdinir/HP-β-CD was prepared using Avicel PH 102 or Ludipress as a direct compression agent and magnesium stearate as a lubricant, achieving a better solubility, stability, and dissolution rate.

### 2.3. Carbapenems

The following description of three carbapenem antibiotics is accompanied by Table 3, summarising the experimental data.

**Meropenem** is characterised by good solubility in water, but also high instability in aqueous solutions due to the hydrolysis of the amide bond in the beta-lactam ring and intermolecular aminolysis resulting in a dimeric product.

Inclusion of meropenem in the β-CD cavity only slightly increased its solubility (by approximately 15%) [96]. However, Paczkowska et al. showed that a meropenem complex with β-CD can serve as a valuable delivery system, significantly contributing to the increased stability of meropenem in aqueous solutions and in the solid phase, keeping the desired meropenem concentration constant over a period of 20 h. The meropenem/β-CD complex exhibited greater bactericidal potency than free meropenem toward *Pseudomonas aeruginosa*, *Rhodococcus equi*, and *Listeria ivanovii* [96].

**Ertapanem** is unstable at room and refrigerated temperatures and is thus prone to degradation with the formation of dimer and open-ring degradation products. To overcome these drawbacks, HP-β-CD was used as an additive in a lyophilised formulation, acting both as a cryoprotective and a stabilising agent [99].

**Tebipenem pivoxil** is the first oral antibiotic in the class of carbapenems. It was complexed with β-CD in a 1:1 molar ratio by the dry mixing method, and characterised by spectroscopic (Fourier-transform infrared spectroscopy (FT-IR) and Raman) and thermal (differential scanning calorimetry (DSC)) methods [100]. As a result, solid-state chemical stability and antibacterial activity against bacterial strains such as *S. aureus*, *P. aeruginosa*, *Enterococcus faecalis*, and *Proteus*
*mirabilis* were increased, but permeability through the Caco-2 cell monolayers was decreased.

## 3. Tetracyclines

Tetracyclines are antibiotics with a very similar structure, theoretically with a wide range of antibacterial activity. Currently, they are significantly limited due to the growing bacterial resistance to antibiotics from this group. Doxycycline is characterised by higher lipophilicity compared to tetracycline, and consequently is better absorbed from the gastrointestinal tract and achieves a higher concentration in many tissues [101]. For details of tetracycline complexation with CD, the reader is referred to Table 4.

An advantage of **doxycycline**/HP-β-CD complexes is increased efficacy against bacterial biofilms, which shows promise in the treatment of aggressive and unresponsive forms of periodontitis [102]. The doxycycline/HP-β-CD decreased the MIC against *Aggregatibacter actinomycetemcomitans* suspension and biofilms 8–16-fold and improved the bactericidal efficacy in comparison with free doxycycline and HP-β-CD complexes at different stoichiometric ratios (1:1, 2:1). In another study, focusing on the adhesion of β-CD to the *S. aureus* cell membrane through hydrogen bonding, a synergistic effect was obtained with the ionic interactions that may arise between the cationic drug and the anionic cell surface, which in turn was responsible for the higher activity of doxycycline/β-CD complex against *S. aureus* compared to pure doxycycline [32]. An in vitro antimicrobial susceptibility test against *S. aureus* revealed a 135-fold decrease in MIC and an 8-fold decrease in minimal bacterial concentration (MBC). Kogawa et al. [103] observed a significant decrease in the rate of photodegradation in aqueous solution after complexation with β-CD: 97% of the doxycycline in complex was unchanged after 6 h exposure to UV radiation, compared with 68% for the pure drug.

On the other hand, complexes of doxycycline hyclate (C_22_H_24_N_2_O_8_·HCl·½ H_2_O·½ C_2_H_5_OH) with different CDs [104] revealed that the photostability of the aqueous solutions of doxycycline/CD complexes can be arranged in the following order: γ-CD > RM-β-CD > HP-β-CD > α-CD, with the best performance shown by doxycycline/γ-CD (1:20 molar ratio) complex. The permeability increased as the inclusions increased. Permeation studies have been performed to investigate the effect of the complexes on biological membrane permeation since doxycycline itself belongs to BCS Class I [104]. NMR studies and molecular modelling allowed to propose a three-dimensional structure of the complex with the phenyl ring included in the CD cavity. Based on 2D-ROESY NMR data, two different arrangements of doxycycline in the cavity of β-CD were proposed, namely the so-called head and tail orientations [105].

For **tetracycline**, the structures and stabilisation energies of β- and γ-CD complexes were calculated using semi-empirical methods (parametrised model 3) and density functional theory (DFT) [111]. A qualitative structure–property relationship was established with two main structural features important for the stabilisation of the IC: (i) inclusion depth, which promotes the hydrophobic contact inside the CD cavity, and (ii) hydrogen bonds established between the guest and the host molecules. In the case of the tetracycline/γ-CD complex, where there is a deeper inclusion than in the case of β-CD, five strong hydrogen bonds can be observed, which increase the stability of this complex.

**Tigecycline** was described in a patent WO 2017/118994, where SBE-β-CD was used as a stabilising agent for a freeze-dried pharmaceutical composition for parenteral administration [115].

## 4. Chloramphenicol and Florfenicol

The experimental conditions and selected results of the cited reports are summarised in Table 5.

**Chloramphenicol** (Figure 1) was introduced into medicine in 1949. Chloramphenicol has a broad spectrum of antibacterial activity and high clinical efficacy; however, it causes severe bone marrow suppression by idiosyncrasy (reaction that occurs at the first contact with the substance that is unpredictable, irreversible, and frequently fatal) with an incidence of 1 case in 24,000 to 40,000 courses of therapy [23]. However, the most common symptom is a predictable dose-dependent toxicity, which usually occurs when the serum concentration of chloramphenicol exceeds 25 mg/L for prolonged periods of time [116].

Chloramphenicol is only used in the cases of severe, life-threatening conditions where there is no alternative to its use. In many countries, chloramphenicol has been withdrawn from systemic preparations and remained in topical formulations only. Chloramphenicol is a lipophilic antibiotic, well-absorbed from the gastrointestinal tract, with a bioavailability of 90–100%, but with poor water solubility due to the presence of the nitrobenzene moiety, and is characterised with a bitter taste. As a result, hydrophilic prodrugs (chloramphenicol sodium succinate or chloramphenicol palmitate) are generally used for the preparation of pharmaceutical compositions. The inclusion into the cavity of the CD molecule can reduce the toxicological properties of chloramphenicol [23]. Chloramphenicol palmitate is sparingly soluble in water (<4.3 × 10^−8^ M) and has several polymorphic forms (A—stable, B and C—metastable) [117]. In the absence of additives, the spray-dried chloramphenicol palmitate was mainly converted to Form subB, which was easily and quickly transformed to metastable Form B. On the other hand, in the presence of HP-β-CD, chloramphenicol palmitate was converted by spray-drying to a stable amorphous complex, which transformed to Form B only in small amounts under severe storage conditions. Dissolution in aqueous medium indicated that the amount of chloramphenicol palmitate released from the chloramphenicol palmitate/HP-β-CD complex was significantly greater than that of the polymorphs (rate and amount: complex >> Form B > Form subB > Form A) [117].

Ali et al. [118] conducted a NMR spectroscopic study of mixtures of β-CD with D-(-)-chloramphenicol, present in two tautomeric forms in solution, and revealed the formation of a 1:1 IC, with the aromatic ring of the guest tightly held by the host cavity. Fatiha et al. [119] carried out computational studies, using semi-empirical methods, to investigate the structures and properties of the ICs of chloramphenicol tautomers with β-CD at 1:1 stoichiometry. Two possible orientations in the host cavity were considered for both enol and keto chloramphenicol. The NMR chemical shifts (in ppm) of the free and complexed chloramphenicol were calculated at the B3LYP/6-31G(d) level of theory and compared with experimental data taken from the literature.

**Table 5 pharmaceutics-14-01389-t005:** Detailed information of chloramphenicol and florfenicol complexes on the types of CDs used, methods of complex preparation, and characterisation techniques *.

API	CD	API:CD Ratio	System	Addition	Preparation	Characterisation	Stability Constant K_c_ (M^−1^)	Solubility Improvement	Ref.
chlorampehnicol	β-CD, HP-β-CD	1:1, 1:2	binary	—	spray-drying	PSS, dissolution, HPLC, UV-VIS, DSC, XRPD, NMR, in vivo studies	120 (HP-β-CD), 170 (β-CD)	PSS ^6^	[117]
	β-CD	1:1	binary	—	in solution	NMR	n/a	n/a	[118]
	β-CD	1:1	binary	—	in silico	quantum mechanical calculations	n/a	n/a	[119]
	β-CD	1:1	binary	—	in solution	resonance Rayleigh scattering, UV-VIS	dependent on the method and temp.	n/a	[120]
	HP-β-CD, M-β-CD	1:1	binary	—	kneading	PSS, DSC, UV-VIS, antimicrobial activity	86.3 (HP-β-CD), 259.5 (M-β-CD)	4.4 → 30.7 (HP-β-CD), 42.0 g/L (M-β-CD)	[121]
	Dimeb	1:1	binary	—	co-evaporation	PSS, UV-VIS, circular dichroism spectroscopy, FT-IR, NMR	493	2.24-fold (in 20 mM Dimeb solution)	[122]
	β-CD, Trimeb	1:1	binary	—	co-dissolution, co-grinding	single-crystal X-ray diffraction, XRPD, TGA, FT-IR, solid-state NMR, SEM, DFT, antibacterial activity	n/a	n/a	[123]
	β-CD	1:1	ternary	glycine, cysteine	freeze-drying	PSS, HPLC, NMR, FT-IR, DSC, TGA, XRPD, SEM, antimicrobial activity, chemiluminescence	180 (binary), 107 (with glycine), 101 (with cysteine)	1.5-fold	[23]
	β-CD	1:1	ternary	NAC	freeze-drying	PSS, HPLC, NMR, XRPD, SEM, FL, chemiluminescence, microbiological studies	75	1.5-fold	[124]
	γ-CD	1:1	ternary	NAC	freeze-drying	PSS, ITC, NMR, SEM, DSC, TGA, XRPD, MM, FL, electron probe microanalysis, antibacterial activity	68 (without NAC), 165 (with NAC)	2.8 → 7.6 (binary), 14.4 mg/mL (with NAC)	[125]
	γ-CD	1:1	NPs	silver NPs	freeze-drying	XRPD , FT-IR, DSC, TGA, SEM, EDX, TEM, NMR, UV-VIS, SERS, MM, zeta potential, antibacterial activity	n/a	n/a	[126]
	SBE-β-CD	1:1	eye drop formulation	PVP, PVA, PVP, HPMC	freeze-drying	PSS, UV-VIS, DSC, XRPD, MM, NMR, SEM, EDS, HPLC-MS/MS, in vitro release, in vitro permeabilization tests, in vivo studies	n/a	PSS ^6^	[127]
florfenicol	HP-β-CD	n/a	binary	—	freeze-drying	SEM, DSC, XRPD, FT-IR, NMR, muscle irritation test, in vivo pharmacokinetic studies	n/a	2.23 → 78.9 mg/mL	[128]
	β-, HP-β-, γ-CD, Captisol	n/a	ternary	PEG-300	freeze-drying	PSS, HPLC	1430 (β-CD), 612 (γ-CD), 817 (HP-β-CD), 1021 (Captisol)	PSS ^6^	[129]
	HP-β-CD	1:1	microparticles	chitosan	evaporation, freeze-drying, spray-drying	PSS, HPLC, DSC, SEM, dissolution	181.4 (without addit.)	PSS ^6^	[130]
	HP-β-CD	n/a	microparticles	PVP, HPMC	as described ^4^	SEM, UV-VIS, HPLC, particle size analysis, stability, in vitro release, pharmacokinetic study antibacterial activity	n/a	n/a	[110]
	γ-CD	n/a	MOF	KOH, CH_3_OH	as described ^4^	SEM, XRPD, UV-VIS, FL, HPLC, in vitro release, antibacterial activity	n/a	9.12 → 76.11 mg/L	[131]

* Abbreviations used in the table are expanded in the section “List of abbreviations”, and additional explanations are provided below Table 1.

Li et al. [120] applied the resonance Rayleigh scattering technique to determine the inclusion constant of chloramphenicol into β-CD, and compared the results with the data obtained using UV-VIS spectroscopy. Based on the temperature-dependence of the stability constant, the thermodynamic parameters (ΔH, ΔS, and ΔG) related to the inclusion process were calculated, revealing that the process is exothermic and enthalpy-driven (|ΔH| > T|ΔS|). Zuorro et al. [121] studied the enhancement of the solubility and antibacterial activity of chloramphenicol after complexation with HP-β-CD and M-β-CD in eye drop formulations. There are no significant differences in the behaviour of chloramphenicol/HP-β-CD and chloramphenicol/M-β-CD, suggesting that the chemical nature of the substituents in the β-CD ring has a limited effect on antimicrobial activity, despite the observed differences in the stability constants (259.5 M^−1^ for M-β-CD versus 86.3 M^−1^ for HP-β-CD). Shi et al. [122] showed that the solubility of chloramphenicol in the presence of heptakis-(2,6-di-*O*-methyl)-β-CD (Dimeb) increased 2.2-fold, and is greater than in the presence of β-CD, HP-β-CD, and M-β-CD, similar to the complex stability constant. The results obtained by spectroscopic methods showed that the nitrophenyl moiety of chloramphenicol was deeply inserted into the cavity of Dimeb from the narrow rim of Dimeb, which differs from the complex with β-CD.

Three different procedures were applied to prepare ICs of chloramphenicol in the solid state: (i) co-crystallisation with β-CD from aqueous solution, (ii) the solvent-free co-grinding with β-CD, and (iii) co-dissolution in ethanol with heptakis-2,3,6-tris-*O*-methyl-β-CD (Trimeb) [123]. The co-crystallisation procedure was optimal for preparing chloramphenicol complexes with β-CD (1:1) and resulted in the formation of microcrystals showing polymorphism. Only the crystals of the latter were suitable for the single-crystal diffraction experiment. However, the data for the guest atoms comprised of a smeared-out electron cloud, so theoretical calculations were applied to propose a reliable geometry and the location inside the host molecule. The co-grinding procedure with β-CD and co-dissolution with Trimeb (1:1) led to obtaining an amorphous material. Both the complexes with β-CD and Trimeb demonstrated selective action against *E. faecalis* strains and a remarkable ten-fold MIC reduction against *Listeria monocytogenes*.

Aiassa et al. [23] improved the solubility of chloramphenicol and reduced the drug-induced production of reactive oxygen species (ROS) in leukocytes, using β-CD complexation with the addition of auxiliary substances, glycine or cysteine. ROS are responsible for damage to the blood-forming organs, which may be related to the potential of chloramphenicol for nitro-reduction and subsequent production of nitric acid, which in turn may cause haematotoxicity in susceptible people [132]. Moreover, it has been shown that the antimicrobial activity of complexed chloramphenicol is retained against *S. aureus*, *E. coli*, and *P. aeruginosa* when tested by agar diffusion methods [23]. Further ternary complexes consisting of chloramphenicol, *N*-acetylcysteine (NAC), β-CD [124], and γ-CD to improve the solubility, antibiofilm activity, and safety profile of chloramphenicol [125] were prepared. It was shown that complexation [124] decreased the biomass and cellular activity of *Staphylococcus* spp., as assessed by the crystal violet and XTT assays. The complexation also resulted in the reduced leukocyte toxicity. One way to induce the resistance to antibiotics is through the formation of a bacterial biofilm [133]. In this context, there is an urgent need to prepare drug-delivery systems responsible for the enhanced inhibition of bacterial biofilm production. NAC as an acetyl derivative of cysteine was investigated as a protective antibiofilm and antioxidant agent [125]. The solubility of chloramphenicol increased 2.7-fold for the binary complex and 5.1-fold for the ternary system. The XTT reduction assay demonstrated a 1.9-fold decrease in the metabolic activity in the case of methicillin-sensitive *S. aureus* and a 1.3-fold decrease in the case of MRSA.

**Florfenicol** (FF) is approved for veterinary use only. Fan et al. [128] conducted physicochemical and in vivo pharmacokinetic studies of FF/HP-β-CD complexes in a water-soluble injection. Compared to the commercial injection of FF, the HP-β-CD complex increased the solubility (35.4-fold compared to FF alone), elimination half-life, transport rate constant, and peak concentration after an intramuscular injection in beagle dogs, as well as shortened the distribution half-life, absorption rate constant, apparent volume of distribution, and peak time. The patent WO 2008/133901 A1 [129] described PSS of FF with the use of α-, β-, γ-, HP-β-CD, Captisol, and PEG-300. It was shown that the synergism of CD solutions with PEG-300 reduced the amount of solvent (PEG) necessary to obtain the required concentration.

## 5. Quinolones

This section discusses the CD complexes of the next generations of quinolone antibacterial agents: the “old” quinolones (first generation) and fluoroquinolones (second–fourth generations). The experimental details are summarised in Table 6.

### 5.1. First-Generation Quinolones

First-generation quinolones were introduced for the treatment of urinary tract infections. Attempts to use them in other infections have failed because these compounds are unlikely to generate adequate concentrations in body fluids other than urine [134]. The side effects and the easy spread of resistance mean that these compounds are not recommended today; therefore, the scope of the available articles was limited.

Shehatta et al. [135] studied the inclusion of **nalidixic acid** into α- and β-CD cavities using UV-VIS spectroscopy and electrochemical methods such as differential pulse stripping voltammetry and cyclic voltammetry. The logarithms of the binding constants were calculated from voltammetric data, indicating that nalidixic acid was bound more strongly to β-CD, with a more apolar cavity, than to α-CD. A combination of the experimental results and molecular modelling studies allowed the structure of the complexes to be proposed, with the 2-methylpyridine group inside the CD cavity and the carboxyl group of the drug remaining outside the β-CD cavity, forming an additional hydrogen bond with the OH group of β-CD. In another study, β-CD/nalidixic acid solid dispersions were prepared, thus achieving an enhanced dissolution rate for these systems compared to solid dispersions with PVP (obtained by solvent evaporation) or sodium starch glycolate (SSG) [136]. The relative potency of the carriers to enhance the dissolution rate of nalidixic acid was in the following order: β-CD > PVP > SSG. Levya et al. [137] used UV-VIS spectroscopy to determine binding constants of γ-CD complexes with nalidixic acid and oxolinic acid under acidic and basic conditions, demonstrating drug penetration into the CD cavity by NMR.

**Table 6 pharmaceutics-14-01389-t006:** Detailed information of quinolone complexes on the types of CDs used, methods of complex preparation, and techniques used to characterise the obtained products *.

API	CD	API:CD Ratio	System	Addition	Preparation	Characterisation	Stability Constant K_c_ (M^−1^)	Solubility Improvement	Ref.
nalidixic acid	α-, β-CD	1:1	binary	—	in solution	PSS, UV-VIS, MM, electrochemistry: differential pulse stripping voltammetry, cyclic voltammetry	398 (α-CD), 1585 (β-CD)	n/a	[135]
	β-CD	n/a	binary	—	kneading	SEM, XRPD, dissolution	n/a	n/a	[136]
	γ-CD	1:1	binary	—	in solution	UV-VIS, NMR	3480 (in 0.03 M HCl), 3760 (in 0.03 M NaOH)	n/a	[137]
piromidic acid	DM-β-CD	1:2	binary	—	neutralisation method	PSS, UV-VIS, DSC, NMR, dissolution	244	PSS ^6^	[138]
	β-CD, DM-β-CD	1:2	binary	—	neutralisation, co-precipitation	PSS, UV-VIS, DSC, NMR, XRPD, dissolution	77.5 (β-CD), 244 (DM-β-CD)	PSS ^6^	[139]
pipemidic acid	β-CD	1:1	binary	—	kneading	PSS, UV-VIS, FT-IR, NMR, MM, bioactivity evaluation	250.8 (pH 4.6), 88.5 (pH 6.8), 86.7 (pH 8.6)	PSS ^6^	[140]
oxolinic acid	HP-β-CD	1:1	binary	—	in solution	UV-VIS, HPLC (photodegradation), NMR	2.65 × 10^3^	n/a	[141]
	γ-CD	1:1	binary	—	in solution	UV-VIS, NMR	1616 (in 0.03 M HCl), 1765 (in 0.03 M NaOH)	n/a	[137]
ofloxacin	β-CD	n/a	binary	—	in solution	solubility studies, DSC, NMR, HPLC (photostability), MM	152 (pH 8.3)	n/a	[142]
	β-CD, HP-β-CD	1:1	binary	—	in solution	UV-VIS, FL, NMR	dependent on the type of CD and pH	n/a	[143]
	β-CD, HP-β-CD	1:1	binary	—	freeze-drying	FT-IR, TGA, DTA, NMR, ITC, DLS, antitumour and antibacterial activity	880 (β-CD), 65.2 (HP-β-CD)	n/a	[144]
	HP-β-CD	1:1	binary	—	kneading	PSS, UV-VIS, FT-IR, MM	1.28 × 10^3^	0.3 → 1.1 mg/mL	[145]
	α-, β-CD	1:1	binary	—	as described ^4^	UV-VIS, FT-IR, NMR, SEM, MM, time-resolved fluorescence	as described ^4^	n/a	[146]
	β-CD	1:1	binary	—	in solution	fluorescence quenching method	1.02 × 10^6^ (neutral pH), 0.99 × 10^6^ (acidic pH)	n/a	[147]
	M-β-CD	1:1	binary	—	co-precipitation	UV-VIS, FL, FT-IR	7.8 × 10^3^	n/a	[148]
	M-β-CD	1:1	binary	—	solid-state	solubility studies, FL, FT-IR, SEM, NMR, MM	167 (pH 3.05), 1000 (pH 7.53), 200 (pH 10.53)	1.28 → 4.26 mg/mL	[149]
	α-CD	1:1	NFs	PEG, PVA	as described ^4^	FT-IR, NMR, SEM, EDX, UV-VIS, antibacterial activity	n/a	n/a	[150]
ciprofloxacin	β-CD	1:1	binary	—	co-precipitation	NMR, FL, IR, DSC, SEM	278	n/a	[151]
	HP-β-CD	1:1	binary	—	co-precipitation	NMR, FL, IR, DSC, SEM	343	n/a	[152]
	α-, β-CD	1:1	binary	—	precipitation	UV-VIS, FT-IR, NMR, SEM, MM, time-resolved fluorescence	as described ^4^	n/a	[146]
	HP-β-CD, M-β-CD	n/a	hydrogel	agar, ethyleneglycol diglycidylether	as described ^4^	FT-IR, UV-VIS, drug release, microbiological tests	n/a	n/a	[153]
	β-CD	n/a	hydrogel	sterculia gum, carbopol 940, *N*,*N*′-methylenebisacrylamide ammoniumpersulphate	as described ^4^	cryo-SEM, FT-IR, solid-state NMR, drug release, UV-VIS, biomedical properties of hydrogels	n/a	n/a	[154]
	poly(cyclodextrin citrate)	n/a	hydrogel	chitosan	as described ^4^	SEM, rheological parameters, degradation studies, drug release, HPLC, antibacterial activity, cytotoxicity	n/a	n/a	[26]
	α-CD	n/a	hydrogel	poloxamer 407	as described ^4^	rheological parameters, in vitro release, HPLC, permeation, antibacterial studies	n/a	n/a	[155]
	HP-β-CD	1:1	eye drop formulation	HPMC, PVP	as described ^4^	PSS, UV-VIS, stability, drug release	175 (pH 5.5), 83 (pH 7.4)	3-fold (pH 5.5), 2-fold (pH 7.4)	[156]
	HP-β-CD	1:1	eye drop formulation	Carbopol 934 and 940, Poloxamer 407 and 188, HPMC	as described ^4^	FT-IR, DSC, NMR	n/a	n/a	[157]
	β-CD	n/a	polyurethane composite	poly(butyleneadipate), 4,4′-diphenylmethane-diisocyanate	as described ^4^	FT-IR, NMR, wide-angle XRD, TGA, DSC, SEM, EDX, antimicrobial activity	n/a	n/a	[158]
	β-CD	n/a	polyurethane composite	HDI, calcium β-glycerophosphate	kneading	FT-IR, XRPD, TGA, UV-VIS, solid-state NMR	n/a	n/a	[159]
	α-, β-CD	1:1	NFs	PCL	solvent evaporation, ultrasonic	solubility study, solid-state NMR, FT-IR, XRPD, SEM, UV-VIS	n/a	PSS ^6^	[160]
	β-CD	1:1	NFs	gelatine	freeze-drying	PSS, UV-VIS, NMR, FT-IR, XRPD, TGA, SEM, MM, dissolution	n/a	PSS ^6^	[161]
	β-CD	n/a	NFs	polylactic acid	co-precipitation	solid-state NMR, SEM, TGA, Raman spectroscopy, UV-VIS	n/a	n/a	[162]
	mono-6-prop- argylamino- 6-deoxy-β-CD	n/a	NFs	azidated cellulose fibres	as described ^4^	FT-IR, SEM, XPS, XRPD, drug-release study, antibacterial assay	n/a	n/a	[163]
ciprofloxacin	β-, γ-CD	n/a	macromolecule	chloroacetyl chloride	as described ^4^	NMR, FT-IR, MS, UV-VIS, drug release, antibacterial activity (MTT assay)	n/a	n/a	[164]
	*O*-p-toluene- sulfonyl-β-CD	n/a	NPs	chitosan, 3-chloro-2-hydroxypropyl trimethyl ammonium chloride	as described ^4^	solubility studies, UV-VIS, DLS, FT-IR, XRPD, SEM, EDX, NMR, antibacterial and anti-fungal activity, in vitro release	n/a	PSS ^6^	[165]
	M-β-CD, poly-M-β-CD	1:1	vascular grafts	PET, citric acid	as described ^4^	solubility study, UV-VIS, NMR, in vitro release	55.9 (M-β-CD), 793.8 (poly-M-β-CD)	PSS ^6^	[166]
	HP-γ-CD	n/a	vascular prosthesis	PET, citric acid	as described ^4^	UV-VIS, SEM, microbiological tests, in vitro cell proliferation	38	n/a	[167]
pefloxacin	α-, β-, HP-β-CD	1:1	binary	—	co-precipitation	NMR, FL	730 (α-CD), 140 (β-CD), 760 (HP-β-CD)	n/a	[168]
lomefloxacin	HP-β-CD	1:1	binary	—	solvent evaporation	FT-IR, XRPD, UV-VIS, in vitro dissolution, in vivo absorption studies, stability	n/a	n/a	[169]
rufloxacin	β-CD, HP-β-CD, γ-CD	1:1	ternary	HPMC	in solution	PSS, UV-VIS, HPLC, bioavailability studies	139 (β-CD), 95 (HP-β-CD), 48 (γ-CD), 111 (HP-β-CD, HPMC, pH 7.4)	PSS ^6^	[170]
norfloxacin B_I_ and C	β-CD	1:1	binary	—	kneading	solid-state NMR, XRPD, FT-IR	n/a	n/a	[171]
norfloxacin A	β-CD	1:1	binary	—	kneading, freeze-drying	PSS, DSC, TGA, FT-IR, XRPD, solid-state NMR, in vitro dissolution, microbiological studies	14 (in water) 58 (pH 6.0) 72 (pH 8.0)	1.2-fold (in water) 1.9-fold (pH 6.0) 2.4-fold (pH 8.0)	[172]
norfloxacin B_I_	β-CD	1:1	binary	—	kneading, freeze-drying	PSS, HPLC (stability), dissolution, microbiological studies	n/a (in water) 40 (pH 6.0) 33 (pH 8.0)	1.2-fold (in water) 1.5-fold (pH 6.0) 1.3-fold (pH 8.0)	[173]
norfloxacin C	β-CD	1:1	binary	—	kneading	PSS, HPLC (stability), dissolution, solid-state NMR, FT-IR, XRPD, SEM	n/a	0.29 → 0.34 mg/mL (in water), decrease at pH 6.0 and 8.0	[174]
norfloxacin	β-CD, HP-β-CD	n/a	binary	—	freeze-drying	PSS, UV-VIS, XRPD, DSC, dissolution	n/a	PSS ^6^	[175]
	β-CD, HP-β-CD	n/a	binary	—	physical trituration, kneading, solvent evaporation	PSS, UV-VIS, in vitro dissolution, SEM, FT-IR, DSC, XRPD	103.5 (β-CD) 642.7 (HP-β-CD)	0.39 → 10.90 mg/mL	[176]
	β-CD	1:1	binary	—	in silico	DFT	n/a	n/a	[177]
	2-methyl-β-CD	1:1	binary	—	in solution	UV-VIS, FL, NMR	2.075 × 10^4^ (pH 3.05) 1.315 × 10^4^ (pH 6.53) 1.425 × 10^3^ (pH 10.53)	n/a	[178]
	β-CD, HP-β-CD, γ-CD	1:1	binary	—	solvent evaporation, co-evaporation, kneading, freeze-drying, spray-drying	PSS, potentiometric titration, DSC, FT-IR, XRPD, SEM, hot-stage microscopy, in vitro dissolution, antimicrobial assay	121.1 (β-CD) 65.9 (HP-β-CD) 84.6 (γ-CD)	up to 2.4-fold	[179]
	β-CD	1:1	binary	—	co-evaporation, kneading followed by freeze-drying	potentiometric titrations, NMR, HPLC (stability), DSC, TGA, FT-IR, XRPD, antimicrobial assay	n/a	up to 2.4-fold	[180]
	β-CD	n/a	ternary	HPMC	solvent evaporation	PSS, UV-VIS, SEM, DSC, FT-IR, XRPD, in vitro dissolution	103.5 (binary system) 253.3 (2.5% *w*/*v* HPMC) 307.5 (5% *w*/*v* HPMC)	0.39 → 4.23 (1:1), up to 6.92 mg/mL (1:1, 5% *w*/*w* HPMC)	[181]
	β-CD	n/a	ternary	citric acid, ascorbic acid	kneading, solvent evaporation	PSS, UV-VIS, IR, DSC, particle size analysis, in vitro dissolution, microbiological studies	22.4	PSS ^6^	[182]
	HP-β-CD	1:1	ternary	glutamic acid, proline, lysine	kneading, freeze-drying	PSS, UV-VIS, NMR, DSC, TGA, in vitro release	n/a	as described ^4^	[183]
	β-CD	1:1	liposome	soybean phospholipids and cholesterol	freeze-drying	FT-IR, XRPD, NMR, MM, TEM	11	n/a	[184]
	β-CD	n/a	nanosponges	diphenyl carbonate	as described ^4^	HPLC, DLS, DSC, FT-IR, TEM, zeta potential, permeability (Ussing chamber experiments), in vitro release, antimicrobial in vivo experiments	n/a	n/a	[185]
	β-CD	n/a	tablet formulation	PVP, HPMC	kneading	PSS, UV-VIS, in vitro dissolution	333	PSS ^6^	[186]
enrofloxacin	α-, β-, γ-, HP-β-CD	1:1	ternary	citric acid	kneading	PSS, UV-VIS, DSC, TGA	20.5 (α-CD), 35.6 (β-CD), 14.0 (γ-CD), 29.5 (HP-β-CD)	255% (α-CD), 38% (β-CD), 232% (γ-CD), 1258% (HP-β-CD)	[187]
	HP-β-CD	1:1	binary	—	as described ^4^	UV-VIS, NMR, FT-IR, HPLC, SEM, dissolution, pharmacokinetic studies	n/a	916-fold	[188]
	γ-CD	n/a	MOF	KOH, CH_3_OH	as described ^4^	SEM, XRPD, UV-VIS, FL, HPLC, in vitro release, antibacterial activity	n/a	158.45 → 372.14 mg/L	[131]
	β-CD	n/a	covalent organic framework	tetraphtaladehyde	as described ^4^	SEM, TEM, FT-IR, HPLC, drug release, cytotoxicity test, antibacterial ability	n/a	n/a	[189]
sparfloxacin	HP-β-CD	1:1	binary	—	as described ^4^	PSS, UV-VIS, FL, FT-IR, potentiometric titration, dissolution	248.8	PSS ^6^	[190]
	α-, β-CD	1:1	binary	—	as described ^4^	UV-VIS, FT-IR, NMR, SEM, MM, time-resolved fluorescence	as described ^4^	n/a	[146]
	β-CD	1:1	binary	—	co-precipitation	FL, NMR, FT-IR, DSC, SEM	0.5 × 10^2^	n/a	[191]
levofloxacin	HP-β-CD	n/a	NPs	chitosan, tripolyphosphate	co-precipitation	PSS, in vitro release, UV-VIS, SEM particle size analysis, zeta potential, accelerated stability studies	n/a	n/a	[192]
	SBE-β-CD	1:1	NPs	chitosan	ionotropic gelation method	UV-VIS, NMR, HPLC (stability), in vitro release, antibacterial activity	n/a	n/a	[193]
	β-CD	1:1	NPs	curdlan, epi- chlorohydrin	freeze-drying	SEM, FT-IR, DLS, in vitro release, antibacterial activity, cell culture studies	n/a	n/a	[194]
	β-CD	n/a	dendrimers	polyamidoamine	as described ^4^	NMR, FT-IR, FL, MM, dialysis experiments	n/a	n/a	[195]
	β-CD	n/a	polypropylene mesh devices	HDI	as described ^4^	SEM, EDX, FT-IR, antibacterial activity, drug release	n/a	n/a	[196]
tosufloxacin tosylate	HP-β-CD	1:1	binary	—	solvent evaporation	PSS, UV-VIS, in vitro dissolution, XRPD, SEM, DSC, FT-IR, NMR	2461	42 times (0.246 → 10.368 mg/mL)	[197]
	HP-β-CD	1:1	binary	—	supercritical antisolvent method	UV-VIS, FT-IR, XRPD, SEM, EDX, dissolution	n/a	n/a	[198]
	HP-β-CD	n/a	binary	—	solution-enhanced dispersion with supercritical CO_2_	UV-VIS, SEM, particle size analysis, DSC, TGA, XRPD, FT-IR, NMR, MM, solubility, in vitro dissolution, antibacterial activity	n/a	6.6 times (up to 489.87 μg/mL)	[199]
moxifloxacin	β-CD	1:1	binary	—	freeze-drying	FL, UV-VIS, FT-IR, SEM, NMR	395	n/a	[200]
	β-CD	1:1	binary	—	freeze-drying	NMR, capillary electrophoresis, MS, FT-IR, DSC, MM, antibacterial activity	324	n/a	[201]
	M-β-CD	n/a	binary	—	in solution	UV-VIS, FT-IR, MM, drug release	2.5 × 10^4^	n/a	[202]
	HP-β-CD	1:1	binary	—	rapid expansion of supercritical solutions	SEM, IR, XRPD, circular dichroism, equilibrium dialysis	n/a	n/a	[203]
	SBE-β-CD and its oligomer	n/a	NPs	HDI	freeze-drying	UV-VIS, FT-IR, NP tracking analysis	10^4^ (SBE-β-CD) 2 × 10^5^ (SBE-β-CD oligomer)	n/a	[204]
	HP-β-CD, M-β-CD, SBE-β-CD oligomers	n/a	NPs	HDI	freeze-drying	UV-VIS, NMR, FT-IR, DLS, equilibrium dialysis, circular dichroism, antibacterial activity, NP tracking analysis	dependent on the type of CD and molar excess of the cross-linking agent	n/a	[205]
	SBE-β-CD	1:1	NPs	HDI	freeze-drying	UV-VIS, NMR, FT-IR, DLS, TEM, equilibrium dialysis, circular dichroism, antibacterial activity, NP tracking analysis	n/a	n/a	[206]
gemifloxacin	HP-β-CD	1:1	binary	—	freeze-drying	FL, UV-VIS, FT-IR, NMR, HPLC/MS, MM	2.7 × 10^2^	n/a	[207]

* Abbreviations used in the table are expanded in the section “List of abbreviations”, and additional explanations are provided below Table 1.

A kneading method was used to prepare β-CD complexes with **pipemidic acid** (1:1) in the solid state [140]. The results showed that the antibacterial activity of this complex against *E. coli* and *S. aureus* was higher than that of the pure drug. Furthermore, tests on human hepatoblastoma HepG2 and MCF-7 cell lines using the MTT assay revealed that the complex exhibited a higher antitumor activity than sole pipemidic acid.

**Oxolinic acid** was studied in combination with HP-β-CD to improve the photostability of this drug [141], demonstrating a 13-fold decrease in the photodegradation rate constant when complexed with HP-β-CD.

### 5.2. Fluoroquinolones

Attaching a fluorine atom in the C6 position of the 1,4-dihydroquinoline ring (Figure 1) results in the formation of the so-called fluoroquinolones. Experimental and computational attempts to classify individual fluoroquinolone drugs into BCS Classes I–IV have been frequently reported [208,209,210].

**Ofloxacin** is characterised by photochemical instability, and hence many articles are concerned with improving drug stability when complexed with CDs. In fact, CD can cover some parts of ofloxacin, which show light sensitivity and are degraded by contact with light, thereby significantly reducing the degradation rate. Koester et al. [142] demonstrated a 2.6-fold enhancement in aqueous solubility upon complexation with β-CD; however, the photodegradation of ofloxacin was not reduced. The authors associated this with the fact that there was only a partial inclusion of the *N*-methylpiperazinyl moiety in the CD cavity, which was confirmed by NMR studies. The piperazinyl ring is probably one of the groups responsible for ofloxacin photodegradation.

Different forms of ofloxacin are known to exist as a function of pH, namely protonated (predominant in acidic media), anionic (predominant in alkaline media), and neutral species. Some studies have suggested that CDs have a different inclusive capacity to different forms of ofloxacin in solutions of different pH levels, namely β-CD is most suitable for the inclusion of the neutral form and HP-β-CD is suitable for the acidic form [143]. The biological activity of β-CD/ofloxacin and HP-β-CD/ofloxacin complexes was presented [144], with particular emphasis on their multifunctional potential as antimicrobial agents against *E.*
*coli* and *S. aureus*, and at the same time as antitumor agents, similar to pipemidic acid [140].

The fluorescence decay curves of ofloxacin, ciprofloxacin, and sparfloxacin with and without CD allowed the lifetimes to be arranged in the following order: ofloxacin/β-CD > ofloxacin/α-CD > ofloxacin > sparfloxacin/β-CD > sparfloxacin/α-CD > sparfloxacin > ciprofloxacin/β-CD > ciprofloxacin/α-CD > ciprofloxacin [146]. The ofloxacin/β-CD complex had the highest binding constant compared to the other compounds tested, suggesting that ofloxacin binds strongly to the β-CD cavity. The interactions of ofloxacin with β-CD were also investigated by the fluorescence quenching method at various temperatures in acidic and neutral medium using copper as a quencher [147]. The effect of temperature on the Stern–Volmer quenching constant and binding constant was analysed, leading to the determination of thermodynamic parameters such as ΔH, ΔS, and ΔG. The negative ΔG value suggested that the binding process is spontaneous in nature due to hydrophobic interactions.

The formation of a complex of M-β-CD with ofloxacin in a 1:1 molar ratio was confirmed by UV-VIS, FT-IR, fluorescence spectroscopy [148], and the NMR method [149]. M-β-CD was more suitable for the inclusion of neutral ofloxacin and the major inclusion interactions between the guest and the M-β-CD cavity were hydrophobic.

**Ciprofloxacin** is one of the best-studied antibacterial drugs in terms of inclusion complexation with CDs. In most cases, the combination of ciprofloxacin with CD is part of a modern form of drug-delivery system, such as a hydrogel, nanofibres, or NPs (described in Section 8). Chao et al. synthesised solid ICs of ciprofloxacin with β-CD [151] and HP-β-CD [152] by the co-precipitation method and proposed their structure on the basis of NMR data.

**Pefloxacin** mesylate was complexed with α-, β-, and HP-β-CD by the co-precipitation method [168]. Based on the 2D NMR spectra, one structure model was proposed for the complexes with α-CD and HP-β-CD, where the bicyclic moiety was deeply included in the CD cavity through the narrow rim, and another model was proposed for the complex with β-CD, involving the shallow inclusion through the wide rim.

An IC of **lomefloxacin** HCl with HP-β-CD (1:1) was obtained by the solvent evaporation method to obtain fast-dissolving tablets to increase patient compliance, increase solubility, and mask the bitter taste [169]. The prepared complex was further compressed into tablets by direct compression using different disintegrants. The best drug release was found in a formulation that additionally contained 1.5% sodium croscarmellose, achieving 100% in 45 min.

**Rufloxacin** was used in combination with CDs and HPMC in the formulation of ophthalmic solutions [170]. The addition of 0.25% (*w*/*v*) HPMC to solutions containing rufloxacin/HP-β-CD complexes increased the solubilising effect of this CD, thus reducing the amount of CD necessary for solubilisation of 0.3% (*w*/*v*) rufloxacin. Preliminary pharmacokinetic data in rabbits indicated that the ocular bioavailability of 0.3% (*w*/*v*) rufloxacin solubilised by HP-β-CD was higher than the 0.3% (*w*/*v*) rufloxacin suspension used as a reference.

**Norfloxacin** is another extensively studied fluoroquinolone drug. Guyot et al. prepared complexes with β-CD and HP-β-CD by the freeze-drying method, which significantly improved the solubility and dissolution rate of the drug [175]. In another study, solvent evaporation was found to be better than physical trituration and kneading in terms of norfloxacin solubilisation, and the HP-β-CD IC with norfloxacin had higher solubility than the β-CD complex when prepared using the same procedure [176]. The dissolution of norfloxacin was below 50%, while the dissolution of norfloxacin/β-CD and norfloxacin/HP-β-CD was more than 80% after 60 min. Norfloxacin was converted from the crystalline to the amorphous form by inclusion complexation.

Mendes et al. [179,180] analysed different techniques of the preparation of norfloxacin ICs with β-CD, HP-β-CD, and γ-CD. The complex of norfloxacin and β-CD (1:1) obtained by kneading followed by freeze-drying led to increased drug solubility, as a result of the amorphous state attributed to the freeze-drying process and the inclusion of norfloxacin into the hydrophobic cavity. The product obtained by the co-precipitation method showed complex formation, although dissolution was slower than that observed with other products due to the formation of a new highly crystalline structure with a larger particle size (confirmed by scanning electron microscopy (SEM) and X-ray powder diffractometry (XRPD)). This method was effective in protecting the drug from photodegradation as well as avoiding hydrolysis. Moreover, the microbiological activity measured by the microorganism growth inhibition zone was enhanced by approximately 23% for the freeze-dried complex [179]. Based on the NMR spectra, the geometric structure of the complex was proposed with the bicyclic moiety inserted into the CD through the wide rim. Norfloxacin/β-CD complex obtained by kneading followed by freeze-drying or spray-drying preserved the antibacterial activity of norfloxacin, as the drug was incorporated into the β-CD cavity, thus protecting it from humidity-induced or thermal degradation to the decarboxylated derivative of no pharmacological action [180].

Several articles describe the complexes of individual polymorphs of norfloxacin with β-CD, obtained by the kneading and freeze-drying methods [171,172,173,174]. Norfloxacin exists in several solid forms: three anhydrous polymorphs (Forms A, B, and C), an amorphous form, a methanol solvate, and several hydrate forms, as well as salts and co-crystals [171]. Norfloxacin A [172], B_I_ (B hydrate) [173], and C [174] were tested separately for complexation with β-CD and their antimicrobial activity, revealing that all forms exhibited microbiological activity, among which the B_I_ form had the most potent activity. This polymorph was found to be the best candidate for the preparation of alternative matrices using β-CD to enhance the biopharmaceutical properties.

Maia et al. [177] performed DFT calculations of the structure and stabilisation energy of the norfloxacin/β-CD complex, thereby revealing that the complex formation was enthalpy-driven and that hydrogen bonds formed between norfloxacin and β-CD played a major role in the complex stabilisation. In addition, theoretical calculations of ^1^H NMR chemical shifts were shown to be an additional procedure to adequately predict the way in which the norfloxacin molecule is incorporated into β-CD. Another study suggested that the main factors affecting molecular recognition were the size match between M-β-CD and the guest, as well as the hydrophobic properties of the guest molecule [178]. The obtained ICs were more stable under acidic conditions, which can be attributed to the hydrophobic effect.

The ternary system showed that the addition of up to 5% (*w*/*w*) of the hydrophilic polymer HPMC improved the solubility of the norfloxacin/β-CD complex (1:1), but a further addition above 5% (*w*/*w*) reduced the solubility of norfloxacin [181]. The solvent evaporation method was used to prepare an IC of norfloxacin/β-CD/HPMC. The complexation with β-CD was shown to significantly improve the dissolution rate of norfloxacin compared to that of norfloxacin alone, with 50% of the drug being released in a quarter of the time. The addition of HPMC did not appear to drastically increase the release of norfloxacin [181]. Dua et al. [182] developed complexes of norfloxacin with β-CD with the addition of ascorbic acid (AA) or citric acid (CA) to lower the pH below 4, where the solubility of norfloxacin improves. The complexes were prepared in 1:1 and 1:2 molar ratios (norfloxacin:β-CD) and with CA or AA in a 1:1:0.5 molar ratio, using the kneading and solvent evaporation method. The results showed an increased dissolution rate at both pH 1.2 and 7.4, with the highest values of 79% and 70%, respectively, being achieved after 120 min for norfloxacin:β-CD:AA complex prepared by solvent evaporation. The results obtained for the in vitro antimicrobial activity of norfloxacin:β-CD:AA against *Bacillus subtilis*, *S. aureus*, and *E. coli* also showed better antimicrobial activity as compared to the pure drug. Other ternary complexes of norfloxacin with HP-β-CD were prepared, using glutamic acid, proline, or lysine as the third component [183]. The highest dissolution rate was obtained with glutamic acid, a negatively charged amino acid. NMR studies showed a partial interaction in the norfloxacin-proline system and the inclusion of lysine inside the HP-β-CD cavity. The latter may suggest that lysine competes with norfloxacin, and hence a lower solubility of the norfloxacin/HP-β-CD/lysine complex was observed.

**Enrofloxacin** is a fluroroquinolone drug approved for veterinary use only. ICs of enrofloxacin with α-, β-, γ-, and HP-β-CD at a 1:1 molar ratio were prepared by the kneading method [187]. The inclusion complex of enrofloxacin with β-CD showed the highest stability constant, but the greatest increase in solubility was obtained using HP-β-CD (HP-β-CD > α-CD > γ-CD > β-CD), namely 1258% [187], or even 916-fold [188]. It was concluded that slight changes in the reaction conditions of the enrofloxacin/HP-β-CD IC formation may be important for its water solubility, and thus for in vivo pharmacokinetic properties such as absorption and bioavailability [188].

**Sparfloxacin** (third-generation fluoroquinolone) is rarely prescribed due to a high incidence of phototoxicity [211]. Mourya et al. [190] determined the influence of temperature on the stability constant of a sparfloxacin/β-CD complex, allowing for the determination of thermodynamic parameters of the inclusion process. The prepared solid-state complex displayed enhanced aqueous solubility and dissolution rate, with 82% drug release from the complex in 20 min, compared with 15% for pure sparfloxacin. In contrast, the measurements of time-resolved fluorescence showed a longer lifetime of sparfloxacin complexes with β-CD than with α-CD [146]. Chao et al. [191] prepared a solid-state sparfloxacin complex with β-CD and proposed its spatial configuration based on 2D NMR studies.

**Tosufloxacin** tosylate was complexed to HP-β-CD by three different methods. The solvent evaporation technique allowed for a 42-fold increase in water solubility, and improved the stability and dissolution of the drug [197]. In the supercritical antisolvent method, the solute was first dissolved in an organic solvent, and then supercritical carbon dioxide was sprayed at high speed and instantaneously diffused into the inside of the solution, resulting in supersaturation of the liquid solution and precipitation of the solute [198]. In phosphate buffer pH 6.8, the dissolution of tosufloxacin tosylate after 105 min was up to 19% for pure API, 35% for the physical mixture with HP-β-CD, and in the range of 52–68% for ICs prepared with different experimental conditions. The method of solution-enhanced dispersion with supercritical CO_2_ utilises the special properties of the supercritical fluid to combine the solubility of the liquid and the diffusion ability of the gas, thereby controlling the speed of supersaturation and consequently the particle size as well as the morphology of the product [199]. Results confirmed the formation of an amorphous IC, resulting in an increase in solubility compared to the pure drug, an enhancement of the dissolution rate from 14% to 61%, and a maintained antibacterial effect of tosufloxacin tosylate against *E. coli* and *S. aureus*.

A complex of **moxifloxacin** (the fourth generation of fluoroquinolones) with β-CD in 1:1 stoichiometry was prepared using the freeze-drying method, and the thermodynamic parameters (ΔH°, ΔS°, and ΔG°) associated with the inclusion process were determined [200]. Szabó et al. [201] continued to investigate the structural properties and antibacterial activity of a moxifloxacin complex with β-CD obtained by the lyophilisation method. The NMR Job plot showed a 1:1 stoichiometry in the liquid state, while ^1^H NMR titrations revealed that the stabilities of the ICs were pH-dependent. The most stable complex was obtained at the pH at which moxifloxacin is present as a neutral molecule (beside monocationic and monoanionic), with the tricyclic moiety entering the host cavity. Studies have shown that supramolecular interactions do not significantly affect the antibacterial activity of the drug. The highest increase of antibacterial activity was found against *E. faecalis*. Kudryashova’s research group published a series of articles on the complexation of moxifloxacin with β-CD derivatives or their oligomers [202,203,204,205,206]. Based on kinetic studies of the moxifloxacin release by equilibrium dialysis, M-β-CD was found to slow down the release of the drug in acidic media by 20–30% compared to the free drug [202]. The following mechanism of complex formation has been proposed: after incorporation of the aromatic fragment of moxifloxacin into the M-β-CD cavity, additional stabilisation of the complex occurs through multiple hydrophobic interactions and hydrogen bonds. Moxifloxacin complexes with HP-β-CD were prepared using a new approach, namely the rapid expansion of supercritical solutions (RESS), using the lyophilisation technique as a reference [203]. However, during lyophilisation, the resulting systems contained residual water, which reduced their stability during storage and greatly limited the ability to use the freeze-drying method. Hence, the RESS method was proposed to overcome this limitation. SEM revealed that particles with a size of 2–4 μm were obtained, which is considerably smaller than for the starting moxifloxacin (15–20 μm). A higher efficiency of drug inclusion in the complex with HP-β-CD was obtained using the RESS technique compared to conventional methods, such as lyophilisation or mixing of solid components. However, unexpectedly, the dissolution rates of the complex obtained by RESS were comparable to un-complexed moxifloxacin over the whole range. On the other hand, the moxifloxacin/HP-β-CD complex obtained by lyophilisation considerably increased the dissolution rate of the drug, which was double in an acidic medium and six times in an alkaline medium.

## 6. Macrolides

Macrolide antibiotics consist of a large, usually 14-, 15-, or 16-membered lactone ring combined with sugar molecules, one of which possesses a dimethylamine moiety. The systematic summary of the experimental details of the cited reports is presented in Table 7.

The **erythromycin** base is sparingly soluble in water and poorly absorbed from the gastrointestinal tract. Moreover, it is unstable, especially in the acidic environment of the stomach, where it undergoes spontaneous cyclisation. Some salts or esters of erythromycin, such as stearate, acistrate, ethyl succinate, propionyl erythromycin lauryl sulphate (estolate), glucoheptonate, and lactobionate, are more stable in acidic environments [228]. Erythromycin and β-CD formed a packing complex (prepared by kneading and solvent evaporation), driven by intermolecular forces, instead of a host–guest structure, due to the limited space in the inner cavity of β-CD [212]. In other words, the drug molecules are too large to be incorporated. Nevertheless, the complex improved the stability of erythromycin in aqueous solution and had a longer duration of bactericidal activity than the free erythromycin. In addition, the complex proved to be non-cytotoxic, and showed significant inhibition of osteoclast formation with simultaneously little effect on osteoblast viability and differentiation. In this case, erythromycin may act as an anti-inflammatory drug at sub-antimicrobial doses, prone to inhibit osteoclast formation.

In contrast to the above-mentioned results, three ICs of erythromycin with β-CD were formed by kneading, co-precipitation, and freeze-drying at a molar ratio of 1:1 [213,214]. The data from FT-IR, XRPD, and DSC showed the IC formation, especially when the co-precipitation and freeze-drying procedures were employed. The highest antimicrobial activity was registered with the complex obtained by co-precipitation, followed by freeze-drying, kneading, and pure erythromycin.

**Clarithromycin** is a semi-synthetic 14-member macrolide, poorly water-soluble (<0.1 mg/mL). Structurally, it differs from erythromycin only in the substitution of the *O*-methyl group for the hydroxyl group in the lactone ring. Salem et al. [217] increased the solubility of clarithromycin at pH 7.4 approximately 700-fold after complexation with β-CD. The efficacy of the β-CD/clarithromycin complex against *Mycobacterium avium* in human peripheral blood monocyte-derived macrophages was slightly lower than that of the free drug, possibly due to the high stability of the IC. In a different approach, the solid-state ternary complexes of clarithromycin with β-CD and citric acid (1:1:1) were prepared by co-evaporation and lyophilisation [218,219] or with polymer Soluplus [220]. The obtained results suggested that the lyophilisation method afforded a higher degree of amorphous unit than co-evaporation, and a part of the guest molecule was located in the β-CD cavity. The dissolution rate at pH 5.0 was similar for pure clarithromycin and its ternary complexes, whereas at pH 6.8 the clarithromycin release was much faster from the complex (about 80% after 40 min) than from the pure drug (about 40% at the same time) [219]. Formulations with a Soluplus concentration of 20% *w*/*w* of the drug content and 30% ethanol as a solvent showed more than 80% of drug release at the end of one hour [220].

Another ternary system, consisting of clarithromycin, HP-β-CD, and PVP K30, was prepared using two methods [221], showing that the spray-dried complex released 91% of the drug within 60 min, while the kneaded complex and pure drug exhibited 71% and 26% drug release within 60 min, respectively. The increase in the dissolution rate of clarithromycin from its ICs may be for several reasons, such as the formation of a soluble complex, amorphisation of the drug, and a reduction of the particle size with consequent improved wettability.

**Spiramycin** consists of a 16-membered lactone ring with 2 amino sugars and 1 neutral sugar. Spiramycin was found to form a non-host–guest complex with M-β-CD in the stoichiometric ratio of 1:3, but not with the native α- or β-CD [223]. The results obtained from the Job plot and FT-IR measurements indicated that the size and geometry of the spiramycin molecule did not allow to be included in the CD cavity, similarly to erythromycin. Interestingly, the authors observed a new effect, namely that the addition of 0.5% (*w*/*v*) of M-β-CD to the culture medium showed a high stimulating effect on the production of spiramycin by *Streptomyces ambofaciens*, while α- or β-CD poorly increased the antibiotic yields.

**Azithromycin** suffers from extensive hydrolytic loss of the cladinose sugar moiety at pH < 6.0, leading to a microbiologically inactive metabolite. The opening of the lactone ring has also been noticed in the pH range of 6.0–7.2. Saita et al. [224] investigated the influence of various native or derivatised CDs on the stability of azithromycin in aqueous medium at pH close to physiological one. The most effective stabilisation of the drug was achieved with SBE-β-CD, which allowed to maintain 99% of azithromycin in the solution for up to 6 months at room temperature. The positive effect of SBE-β-CD was mainly due to the inhibition of the degradation pathway leading to the opening of the lactone ring of azithromycin. In another approach, the effect of different CDs’ (β-CD, epichlorohydrin-β-CD, SBE-β-CD) drug:CD molar ratio (1:1 and 1:2 *w*/*w*) on azithromycin dihydrate solubility was investigated [225]. Depending on the CD used, drug release from the complex decreased in the order: SBE-β-CD > epichlorohydrin-β-CD > β-CD. In vitro drug-release studies for ICs showed that the freeze-drying technique allowed for a better drug release compared to the solvent evaporation method. X-ray diffraction patterns showed a partial transition of the crystalline form of the drug to an amorphous one, based on the loss of the characteristic 2θ peaks of the azithromycin dihydrate.

## 7. Other Antibacterial Drugs

This section combines and discusses antibacterial drugs belonging to the groups of aminoglycosides, glycopeptides, polypeptides, nitroimidazoles, oxazolidinones, as well as three other compounds (fusidic acid, furazolidone, and novobiocin), because only a few reports about CD complexation for each group can be found. Table 8 summarises the experimental conditions of all the works cited in this section.

### 7.1. Aminoglycosides

Aminoglycosides (Figure 1) are large, highly basic, hydrophilic molecules, and hence they are well-soluble in water, but hardly diffuse through cell membranes and thus must be administered parenterally [229]. Most aminoglycosides have a pK_a_ value above 8, so they are largely dissociated in all body fluids, which is important for their behaviour in the body. Several modern delivery systems were designed for these antibiotics, discussed in detail in Section 8.

### 7.2. Glycopeptide and Polypeptide Antibiotics

Glycopeptides are large-molecule antibiotics composed of a peptide ring linked to two sugar molecules (Figure 1). Two glycopeptide antibiotics are currently used in medicine: vancomycin and teicoplanin. Vancomycin is poorly absorbed from the gastrointestinal tract and thus it is administered intravenously in systemic infections. Teicoplanin is structurally related to vancomycin, but its greater lipophilicity results in a better tissue distribution and a longer half-life. The pharmacokinetics of teicoplanin is similar to that of vancomycin. The drug is not absorbed after oral administration but it can be injected intramuscularly [230].

**Table 8 pharmaceutics-14-01389-t008:** Detailed information of complexes of other antibacterial drugs on the types of CDs used, methods of complex preparation, and characterisation techniques *.

API	CD	API:CD Ratio	System	Addition	Preparation	Characterisation	Stability Constant K_c_ (M^−1^)	Solubility Improvement	Ref.
gentamicin sulphate	β-CD	n/a	NFs	polyurethane, nanochitosan	electrospinning	SEM, FT-IR, TGA, tensile strength, contact angle, vapour transmission rate, sorption, in vitro degradation, haemolysis assay, in vitro cytotoxicity, cell proliferation, in vitro release, antibacterial activity	n/a	n/a	[231]
amikacin sulphate	β-CD	n/a	microspheres	terephtaloyle chloride, sorbitan 85 trioleate	freeze-drying	laser diffraction, FT-IR, DSC	n/a	n/a	[232]
paromomycin	HP-β-CD	n/a	microspheres	glyceryl monostearate, soya lecithin, PVA, PEG 400, Tween 80, trehalose	emulsion/solvent evaporation + freeze-drying	particle size analysis, zeta potential, UV-VIS, FL, SEM, TEM, FT-IR, XRPD, in vitro release, in vitro cytotoxicity, in vitro and in vivo antileishmanial activity, in vivo toxicity study	n/a	n/a	[233]
vancomycin	triacetyl-α-CD, triacetyl-β-CD, triacetyl-γ-CD	1:1	binary	—	kneading, co-evaporation, spray-drying	DSC, TGA, particle size analysis, in vitro release, HPLC, antimicrobial activity	n/a	n/a	[234]
	β-CD	1:1	binary	—	freeze-drying kneading	XRPD, TGA, SEM, FT-IR, dissolution, HPLC	n/a	n/a	[235]
	β-CD	n/a	binary	—	in solution	cytotoxicity	n/a	n/a	[236]
	β-CD	n/a	hydrogel	2-isocyanatoethyl 2,6-diisocyanatohexanoate, 1,6-diisocyanatohexane	ss described ^4^	FT-IR, XRPD, SEM, drug release, antibacterial activity	n/a	n/a	[237]
	β-CD	n/a	pseudopolyrotaxane	PEG diglycidyl ether	ss described ^4^	NMR, DSC, XRPD, SEM, rheological properties, in vitro release, UV-VIS, antibacterial activity, cell adhesion and proliferation	n/a	n/a	[238]
	β-CD	1:1	supramolecular amphiphile	oleyamine	ss described ^4^	FT-IR, NMR, DLS, TEM, DSC, MM, in vitro release, antibacterial activity, in vitro cytotoxicity assay	n/a	n/a	[239]
	α-CD	n/a	supramolecular gel	Pluronic F127	ss described ^4^	π-A isotherms, XRPD, FT-IR, rheological properties, release study, antibacterial activity	n/a	n/a	[240]
vancomycin + ceftazidime	HP-β-CD, HP-γ-CD	1:1	two drugs + CD	—	in solution	HPLC, NMR, UV-VIS, turbidimetry, solubility, microbiological studies	n/a	PSS ^6^	[77]
teicoplanin	triacetyl-α-CD, triacetyl-β-CD, triacetyl-γ-CD	n/a	binary	hydroacetonic solution	kneading, evaporative crystallisation under microwave irradiation	DSC, TGA, dissolution	n/a	n/a	[241]
polymyxin B	β-CD	1:1	binary	—	in solution	surface tension measurement	3.0 × 10^3^	n/a	[242]
metronidazole benzoate	β-CD	1:1	binary	—	as described ^4^	PSS, stability, HPLC, UV-VIS, DSC	1.3 × 10^3^	PSS ^6^	[243]
	β-CD	1:1	binary	—	in solution	PSS, TLC, NMR, UV-VIS	251	0.14 → 1.39 mg/mL	[244]
metronidazole	β-CD, low methylated-β-CD (CRYSMEB), RM-β-CD	1:1	binary	—	in solution	PSS, HPLC, antibacterial activity	n/a	1.02 (β-CD) 2.14 (CRYSMEB) 3.65 (RM-β-CD)	[245]
	HP-β-CD	n/a	NFs	—	electrospinning	PSS, rheological properties, SEM, NMR, FT-IR, DSC, TGA, XRPD, in vitro dissolution tests	30	~2 times (in 120 mM HP-β-CD)	[246]
	β-CD, HP-β-CD	1:1	NPs	chitosan	freeze-drying	PSS, UV-VIS, NMR, FT-IR, SEM, DSC, TGA, differential thermogravimetric analysis, in vitro release, antibacterial activity	n/a	PSS ^6^	[247]
	hexanoyl-β-CD ester	n/a	nanospheres	Pluronic PE/F68	as described ^4^	particle size analysis, drug loading, HPLC, in vitro release	n/a	n/a	[248]
metronidazole, ornidazole, tinidazole, secnidazole	β-CD, HP-β-CD, M-β-CD	1:1	ternary	PVP	in solution	solubility studies, calorimetry, NMR	dependent on the drug, type of CD and addition of PVP		[249]
tinidazole	β-CD	1:0.5	formulation	PEG-6000, urea, PVP, gum acacia	solvent-free, microwave-assisted	solubility studies, UV-VIS (stability), FT-IR, XRPD, DSC, MM, microscopy, in vitro release	n/a	3.76 → 36.89 mg/L	[250]
ornidazole	β-CD	n/a	microspheres	epichlorohydrin	as described ^4^	UV-VIS, SEM, FT-IR, NMR, solid state NMR, XRPD, DSC, TGA, MM, in vitro release, antibacterial test	n/a	n/a	[251]
linezolid	β-CD	1:1	binary	—	solvent evaporation	UV-VIS, FL, NMR, MM	351	n/a	[252]
	β-CD	1:1	ternary	HPMC e5LV	kneading, co-evaporation, microwave method	PSS, UV-VIS, XRPD, DSC, FT-IR, MM, in vitro dissolution	1024 (binary), 1393 (5% *w*/*v* HPMC)	PSS ^6^	[253]
tedizolid	β-CD, γ-CD	1:1, 1:2	binary	—	in solution	NMR	dependent on CD and technique	n/a	[254]
	HP-β-CD	1:1	binary	—	kneading	DSC, XRPD, FT-IR, dissolution, HPLC, MM, permeability, antibacterial activity	n/a	n/a	[255]
fusidic acid sodium salt	β-CD, γ-CD	1:2, 1:1	binary	—	in solution	NMR	3.5 × 10^3^ (1:1)	n/a	[256]
fusidic acid	β-CD	1:1	binary	—	kneading co-precipitation freeze-drying	FT-IR, XRPD, SEM, thermal analysis, antimicrobial test	n/a	n/a	[257]
furazolidone	β-CD	n/a	binary	—	kneading, freeze-drying	PSS, UV-VIS, SEM, FT-IR, DSC, TGA, NMR, biological in vitro assays	220	1.68-fold in 12 mM β-CD	[258]
	β-CD, HP-β-CD	n/a	binary	—	kneading, freeze-drying	DSC, TGA, SEM. XRPD, solid-state NMR, MM, Raman chemical imaging, cytotoxic, antibacterial activity	n/a	n/a	[259]
novobiocin	β-CD	n/a	hydrogel	2-isocyanatoethyl 2,6-diisocyanatohexanoate, 1,6-diisocyanatohexane	as described ^4^	FT-IR, XRPD, SEM, drug release, antibacterial activity	n/a	n/a	[237]
	β-CD	n/a	pseudo- polyrotaxane	PEG diglycidyl ether	as described ^4^	NMR, DSC, XRPD, SEM, rheological properties, in vitro release, UV-VIS, antibacterial activity, cell adhesion and proliferation	n/a	n/a	[238]

* Abbreviations used in the table are expanded in the section “List of abbreviations”, and additional explanations are provided below Table 1.

The ability of **vancomycin** to interact with three hydrophobic CDs (triacetyl-α-, -β-, or -γ-CD) and the influence of the preparation technique on the prolongation of the drug delivery for site-specific treatment of bone infections were investigated [234]. It has been proven that the binary systems, prepared by the three methods, kneading, co-evaporation, and spray-drying, have a particle size suitable for parenteral administration at a specific site and are characterised by drug-loading efficiencies close to 100%. All the physical mixtures were characterised by slowing down the vancomycin release, with no marked differences in release profiles as a function of the type of CD used. In the case of the kneading process, a further decrease in drug release was observed, the greatest for the complex with triacetyl-β-CD (only 70% of drug released within 4 h). This result supported the hypothesis, previously made from DSC measurements, that the mechanical stress exerted by kneading can induce an interaction between vancomycin and triacetyl-β-CD. A decrease in the drug-release rate was also evident for the co-evaporation method, where the complex with triacetyl-α-CD was characterised by the lowest drug release (60%, 4 h). The results of spray-drying from the aqueous suspension indicated no drug–CD interaction, confirmed by DSC, while the spray-drying from hydroacetonic solution resulted in the formation of the amorphous products with a consequent decrease of drug release, especially for the system based on triacetyl-γ-CD. Vancomycin binary complexes with β-CD by kneading and freeze-drying were also developed [235]. The in vitro dissolution profiles of vancomycin were determined in a simulated cerebrospinal fluid and showed that a modified release was achieved with the improved vancomycin bioavailability. The kneading method was found to be more effective in prolonging the delivery of vancomycin compared to the freeze-drying. Only 80% vancomycin was delivered from the kneaded complexes after 3 h, whereas for the freeze-dried system, the vancomycin delivery was approximately >90% over the same time period. Then, Zarif et al. [236] supplemented the study of vancomycin/β-CD complexes with an investigation of the cytotoxic effects of the complex on the human glial cell line.

**Teicoplanin**, like vancomycin, has been combined with triacetyl-α-, -β-, or -γ-CD to prolong the release of this hydrophilic drug [241]. Triacetyl-γ-CD slowed down the release of the drug the most and, unexpectedly, the physical mixture preparation technique was found to be more effective in delaying the drug release than kneading and evaporative crystallisation under microwave irradiation.

**Polymyxin B** is an amphiphilic polypeptide antibiotic, as it consists of a ring of amino acids linked by peptide bonds and an additional peptide chain with a different amino acids sequence. Polymyxins exhibit considerable toxicity. This group includes five antibiotics, polymyxins A–E. Two compounds are on the market: polymyxin B, which is applied topically, and polymyxin E, known as colistin [260]. Angelova et al. [242] used the surface tension measurement to determine the association constant of polymyxin B/CD complexes. The ability of CD to form drug/CD ICs decreased in the order: β-CD > α-CD > γ-CD, indicating the importance of steric alignment factors in the formation of host–guest ICs.

### 7.3. Nitroimidazole Derivatives

**Metronidazole** benzoate in combination with β-CD was first studied in 1984 by Andersen and Bundgaard [243]. In addition to increasing the physical stability of the metronidazole benzoate suspension after complexation with β-CD, the inclusion protected the drug from the photochemical degradation and decreased the hydrolysis rate constant by a factor of four. Another study showed that the solubility of benzoyl metronidazole increased 9.7-fold due to the formation of 1:1 benzoyl metronidazole/β-CD complexes in water [244]. Malli et al. [245] used two methylated β-CDs, namely low-methylated β-CD (CRYSMEB) and RM-β-CD, to increase the apparent solubility of metronidazole in water. RM-β-CD, however, is not adapted to the parenteral route as a high degree of methyl substitution is associated with higher toxicity, while CRYSMEB, due to low methyl substitution and partially crystallised CD, shows lower cytotoxicity than RM-β-CD. RM-β-CD allowed for the maximum increase of solubility of metronidazole from 9.2 to 34.3 mg/mL. Both formulations retained activity against *Trichomonas vaginalis*. Chadha et al. [249] investigated the thermodynamics of complexation of 5-nitroimidazoles (including metronidazole) with β-CD, M-β-CD, and HP-β-CD in water and 0.25% PVP using the calorimetric method. The role of the soluble polymers is to increase the complexation efficiency of the drug and to reduce the formulation bulk of solid oral dosage forms. The complex equilibrium constants increased in the order metronidazole < ornidazole < tinidazole < secnidazole and were significantly enhanced by β-CD methylation or the presence of PVP.

Microwave energy was utilised to prepare solvent-free solid dispersions containing **tinidazole**/β-CD complexes [250]. The microwave-assisted method was used to change the crystalline state of the drug to an amorphous state, thus improving its dissolution rate.

### 7.4. Oxazolidinones

Oxazolidinones (Figure 1) are a novel class of fully synthetic antibacterial drugs, among which only linezolid and tedizolid are approved for medical use.

The **linezolid**/β-CD complex was investigated with respect to the drug/β-CD binding to bovine serum albumin (BSA) by means of the measurement of the fluorescence quenching and Förster resonance energy transfer to understand the pharmacokinetics of the drug in β-CD encapsulated form [252]. The stoichiometry of the linezolid/β-CD IC was 1:1, and the binding constant was 3.5 × 10^2^ M^−1^. NMR studies showed that the amide substituent on the oxazolidinone ring of linezolid was involved in its binding to β-CD. Meanwhile, the encapsulation of linezolid by β-CD decreased the strength of binding of the linezolid to BSA by blocking the hydrogen bonding and hydrophobic interaction of the linezolid. Mohapatra et al. [253] investigated the effect of the semi-synthetic, low-viscosity hydrophilic polymer HPMC (2.5% and 5% *w*/*v*) on the increase in solubility and masking the taste of linezolid in the multi-component β-CD IC. The complexes were obtained by different methods with or without HPMC, and the microwave irradiation allowed for obtaining the best results. The drug content, solubility, and dissolution of linezolid in ternary complexes were significantly higher than those of the binary complex.

**Tedizolid** has very low solubility in water, and therefore its prodrugs such as tedizolid phosphate (TED-PO_4_) and tedizolid phosphate sodium salt were developed. Bednarek et al. [254] performed a comprehensive NMR investigation of (R)-tedizolid and its phosphate prodrug in combination with β-CD, γ-CD, as well as heptakis-(2,3-di-*O*-acetyl-6-sulfo)-β-CD. Stoichiometry of the complexes was determined using the Job plot only for TED-PO_4_, showing a 1:2 ratio in the case of γ-CD and predomination of a 1:1 complex with β-CD. T-ROESY spectra allowed to conclude that parent tedizolid penetrates deeper into β-CD and γ-CD cavities than TED-PO_4_. Paczkowska-Walendowska et al. [255] prepared and characterised the solid binary system of tedizolid with HP-β-CD. The increase in the dissolution rate was observed in the presence of HP-β-CD, while maintaining a high permeation coefficient (higher than 1 × 10^−6^ cm/s using the PAMPA system) and high microbiological activity expressed by a decrease in the MIC value in the case of *E. faecalis* and *E. faecium*.

### 7.5. Fusidic Acid and Furazolidone

**Fusidic acid** in the form of a sodium salt formed 1:1 complexes with γ-CD and 1:2 complexes with β-CD, and the structures of the obtained complexes were proposed [256]. Marian et al. [257] used three techniques to prepare solid complexes of fusidic acid with β-CD (1:1). It was confirmed that all the tested compounds showed antimicrobial activity, and the complex obtained with the use of freeze-drying allowed to obtain similar activity against *S. aureus* to pure fusidic acid.

**Furazolidone** belongs to the class of nitrofuran derivatives. Carvalho et al. [258] prepared furazolidone ICs with β-CD by two different techniques to increase drug solubility and reduce drug toxicity associated with high doses. The solubility of the complexes was 1.68 times higher than that of pure furazolidone. The results indicated that β-CD complexes may be a cost-effective alternative for the pharmacotherapy of leishmaniasis in dogs infected with *Leishmania amazonensis*. In another report, the scope of the study was extended to the use of HP-β-CD, and the formation of more effective interactions was observed for a complex prepared by lyophilisation in a 1:2 molar ratio than in a 1:1 ratio or using the kneading method [259].

## 8. Modern Drug-Delivery Systems

### 8.1. Lipid-Based Nanocarriers

Lipid-based nanocarriers can be used for targeted delivery, be administrated by a variety of routes, and even load both lipophilic and hydrophilic drugs [261]. Three drugs, ceftazidime, norfloxacin, and paromomycin, were loaded into liposomes [76,184] and solid-lipid NPs [233]. HP-β-CD with ceftazidime in a molar ratio of 1:1, 1:2, and 1:5 indicated that the liposomes with HP-β-CD with stoichiometry of 1:5 were less stable [76]. The use of HP-β-CD significantly increased the stability of **ceftazidime** and accelerated the drug-release profiles without any significant changes in the release pattern. The release profile of ceftazidime in the initial period up to 1 h and also in the second stage was fitted with Korsmeyer–Peppas kinetics, which was consistent with non-Fickian transport, influenced by diffusion and relaxation [76]. A ternary IC containing the freeze-dried **norfloxacin**/β-CD complex incorporated into a liposome using free multilamellar vesicles was prepared. NMR spectroscopy provided strong evidence of the incorporation of this complex into the liposomes [184]. Whereas, solid-lipid NPs modified with HP-β-CD, loaded with amphotericin B and **paromomycin**, were developed for the treatment of visceral leishmaniasis. In vitro drug release followed a biphasic pattern, showing a sustained drug-release profile of up to 57% (amphotericin B) and 21.5% (paromomycin) within 72 h and an initial burst release (17% amphotericin B, 20% paromomycin) within 4 h as a result of the presence of the drug on the surface of the NPs. [233]. The authors concluded that release occurs mainly by spreading across the matrix of the lipid or/and biodegradation, as well as degradation of the matrix surface.

### 8.2. Polymeric Nanocarriers

#### 8.2.1. Natural Polymer-Based

Chitosan is a biocompatible and biodegradable polymer, widely used in NPs for drug delivery. Three methods of complex preparation were used to obtain a ternary system of florfenicol (FF) with HP-β-CD and chitosan NPs [130]. In vitro studies showed that the FF solubility almost doubled, and a better dissolution profile was exhibited by the product prepared by spray-drying. Drug release from FF microparticles was analysed in HCl of pH 1.2 (simulated gastric media), showing a better profile with dissolution greater than 80% after 15 min for FF/HP-β-CD/chitosan compared to FF alone (less than 10%) and FF/chitosan microparticles (not higher than 50%). The authors postulated that the retardant effect of chitosan could be explained by the slow diffusion of the drug through the more hydrophilic chitosan/HP-β-CD matrix layer around the lipophilic drug [130].

Chitosan NPs based on SBE-β-CD were obtained by the ionotropic gelation method, for the delivery of **levofloxacin** to the eye [193]. The in vitro release profile of levofloxacin was biphasic, showing a burst effect of about 20% in the first hour followed by a sustained release over 72 h (almost 100%). The burst effect may be caused by the desorption of levofloxacin superficially adsorbed on the NPs and/or the rapid diffusion of the drug encapsulated near the NPs’ surface. The release profile showed the best correlation with the Higuchi model (the amount of levofloxacin released from the matrix is proportional to the square root of time), highlighting that the release process is based on Fickian diffusion. Moreover, according to the authors, a positive zeta potential value may favour the interaction of the NPs with the negatively charged ocular tissue, increasing their residence time, and consequently improving the efficacy of the levofloxacin. In vitro antibacterial activity against Gram-positive and Gram-negative bacteria showed that the activity of chitosan/SBE-β-CD NPs loaded with levofloxacin was twice as high as that of the free drug [193].

Similarly, ionic gelation methods have been used for **metronidazole**/β-CD and metronidazole/HP-β-CD ICs that have been embedded in chitosan NPs [247]. Association constants and thermodynamic parameters confirmed the incorporation of metronidazole during complexation with both CDs. The in vitro release profile showed that after 10 min, 89% of metronidazole was dissolved from HP-β-CD ICs, whereas slightly less from β-CD ICs, 81%, and only 47% from pure metronidazole. The antibacterial activity of metronidazole increased after its encapsulation in the CD cavity due to the better diffusion of the drug to the target site. The percentage of inhibition doubled against *Salmonella*, and increased by 25% against *S. aureus*, by 9% against *E. coli*, and by 8% against *Bacillus cereus*, respectively [247].

For **levofloxacin**/HP-β-CD complexes, the liquid/liquid co-precipitation method and the ionic gelation technique were used to formulate NPs consisting of chitosan in the range of 0.1–0.3% (positively charged) and tripolyphosphate in the range of 0.1–0.5% (negatively charged) as the cross-linking agents [192]. Complexation with HP-β-CD (0–25 mM) allowed the sensitive drug to be protected, while the in vitro release profile was characterised by an erratic drug release initially, followed by a delayed release phase. The maximum in vitro release was observed for the formulation without HP-β-CD, achieving 93% over a period of 12 h. All other HP-β-CD formulations showed 80% release at the end of 12 h. The authors concluded that all the formulations were best-fitted to super Case II transport, based on the Korsmeyer–Peppas equation, and that drug release could be due to increased plasticisation at the relaxing boundary.

Dhiman and Bhatia [165] prepared quaternised CD-grafted chitosan (Qβ-CD-g-CH) NPs entrapping **ciprofloxacin** for the development of a sustained release system, as demonstrated during a 24 h study. Qβ-CD-g-CH-ciprofloxacin NPs showed a better inhibitory effect against *E. coli* and *S. aureus* than Qβ-CD-g-CH and chitosan alone. The authors postulated that the mechanism of antibacterial activity was related to the positive charge density of quaternised chitosan absorbed onto the negatively charged cell surface of bacteria, leading to leakage of protein components and other intercellular constituents.

Two anti-tuberculosis drugs, namely rifampicin and **levofloxacin**, were complexed with β-CD, and conjugated to NPs of curdlan (linear β-1,3 glucan, a high molecular weight polymer of glucose) using epichlorohydrin [194], to achieve the simultaneous sustained release of both drugs over an extended period of time. The drug loading on the curdlan NPs conjugating CD ICs was four and five times higher than the direct loading with rimfapicin and levofloxacin, respectively. Rifampicin was released in a sustained and controlled manner up to 72 h, while levofloxacin showed an initial burst release, with 80% released within the first 8 h, when loaded alone, and a sustained release in the dual drug-loaded sample. The release profile showed the best correlation with the Weibull model, and when drug release reached 60%, the profile showed correlation to the Korsemeyer–Peppas model, suggesting that they are consistent with Fickian diffusion. The prepared systems were able to kill more than 95% of *Mycobacterium smegmatis* residing in macrophages within 4 h, thus achieving high therapeutic efficacy.

#### 8.2.2. Synthetic Polymer-Based

Due to the enormous amounts of monomers available and the virtually unlimited ability to functionalise them, synthetic polymers have recently gained importance in the context of drug delivery. **Meropenem**/γ-CD complex prepared by means of the liquid CO_2_ method was encapsulated into poly(lactic-co-glycolic acid) (PLGA) NPs by the double emulsion solvent evaporation method [98]. The following parameters were assessed: drug loading, entrapment efficiency, in vitro release study, apparent permeability coefficient (using the Caco-2 cell monolayer assay), and secretory transport for meropenem, meropenem/γ-CD, and meropenem/γ-CD/PLGA. The release profile of meropenem in universal buffers (gastric pH for 2 h, intestinal pH for 6 h) was 5.1% and 7.6% at simulated gastric pH, while 23.6% and 27.4% at pH 6.8 for active and total drug release, respectively. The resulting multi-material delivery system protected meropenem from gastric pH, making it possible to administer the drug orally with a controlled release while retaining its antibacterial activity (against *S. aureus* and *P. aeruginosa*), but was unable to improve drug permeability or reduce drug efflux rates in a Caco-2 cell monolayer model [98].

Similarly, **roxithromycin** was encapsulated in the β-CD and HP-β-CD cavity also using solvent evaporation, and then each of the resulting complexes was separately loaded into PLGA to synthesise NPs [215]. The designed formulations showed significant activity against the selected multidrug-resistant bacterial strains (i.e., *E. coli* and *S. aureus*) in the following order: roxithromycin/PLGA > HP-β-CD/roxithromycin/PLGA > β-CD/roxithromycin/PLGA. The authors suggested that the difference in antimicrobial properties is mainly due to the formation of strong electrostatic interactions between the individual components of β-CD/roxithromycin/PLGA and HP-β-CD/roxithromycin/PLGA systems, which hinder the release of the drug from these formulations.

A microparticle suspension, which consisted of **doxycycline** HCl, **FF**, and HP-β-CD as a host molecule, PVP as a polymeric carrier, and HPMC as a suspending agent, was prepared by saturated solution stirring combined with the high-pressure homogenisation method [110]. The results demonstrated that the antibiotics showed synergistic or additive antibacterial activity against the causative agents of porcine bacterial pneumonia, such as *Streptococcus suis*, *Actinobacillus pleuropneumoniae*, and *Haemophilus parasuis*, as well as good physicochemical and pharmacokinetic properties. The suspension increased the bioavailability of doxycycline HCl and FF by 1.74- and 1.13-fold, respectively. The suspension increased the stability of doxycycline HCl in aqueous solution while prolonging the release of both compounds in phosphate-buffered saline (PBS) at pH 7.4. After 24 h, the cumulative release was greater than 90% for both drugs, with a burst release within 2 h.

### 8.3. Polymeric Nanosystems Based on CDs

Self-assembly CD-based NPs formed by CMC and a quaternary amino β-CD polymer (QA-β-CD) were prepared for the delivery of **meropenem** [97]. It was assumed that the positively charged QA-β-CD polymer interacted with meropenem, while the negatively charged CMC stabilised the NPs in some way. NMR results showed non-specific interactions, including electrostatic attraction between meropenem and the CD, with shifts being more significant at higher QA-β-CD concentrations. The kinetic studies of drug stability showed a slowing of the hydrolysis of meropenem. Incorporation of meropenem into NPs increased the drug permeation through a single layer of a semi-permeable cellulose ester membrane. The authors suggested that the aggregation of NPs may explain the reduced permeability at higher NPs concentrations by occluding the pores of the membrane [97].

In another work, NPs of SBE-β-CD oligomers were synthesised using 1,6-hexamethyl diisocyanate (HDI) as a bifunctional condensing agent, thus increasing the complexation efficiency of **moxifloxacin** by 20 times. The binding efficiency increased due to the multi-point interaction of the moxifloxacin molecule with the functional groups of the oligomeric carrier [204].

Another paper demonstrated that cross-linking of derivatised CDs (M-, HP-, and SBE-β-CD) with HDI allowed to obtain NPs with diameters of 100–200 nm with distinct binding properties, strongly depending on the nature of the CD substituent, degree of oligomerisation, and charge of the NP [205]. Interestingly, the methyl substituent (hydrophobic), which most improved the binding of moxifloxacin to monomeric CD complexes, had the opposite effect on the CD oligomers. Meanwhile, the SBE substituent (negatively charged), which had only a limited effect on monomeric CD, improved the binding of cross-linked CD by almost two orders of magnitude, whereas the HP substituent (more hydrophilic than the previously mentioned substituents) had a much weaker effect. About 90% of free moxifloxacin was released from the dialysis membrane in less than 45 min, while the use of both systems (SBE-β-CD itself and its oligomers) led to a reduction in the release kinetics of moxifloxacin at the initial section of the curve. With the increased molar excess of cross-linking agent from 1 to 5 and the decreased particle charge, the decrease in the moxifloxacin release was more pronounced. The best results for the antibacterial efficiency against *E. coli* were observed for the sample S5 (the most densely cross-linked of SBE-β-CD oligomers) with up to three-fold higher as compared to free moxifloxacin and the complex with monomer SBE-β-CD. Moreover, the sample S5 showed a tendency for prolonged action [205]. Additionally, SBE-β-CD cross-linked in the presence of moxifloxacin formed polymeric NPs of 50–150 nm in diameter, with a highly efficient (up to 85%) encapsulation of the drug [206]. In the case of highly cross-linked complexes, a substantial part of the moxifloxacin molecule can be captured by the polymer network, rather than absorbed by the CD cavity. This can be accompanied by dramatic changes in the release profile of moxifloxacin: depending on the pH, between 30% (pH 4.0, moxifloxacin is positively charged, the strongest interactions between polymer matrix and moxifloxacin were observed) and 100% (pH 7.4, moxifloxacin is deprotonated) of the drug was released from the polymer network within a week.

The idea of targeting **amikacin sulphate** to the lungs by encapsulation into β-CD (7.5% *w*/*v*) microspheres has been proposed to optimise the therapeutic efficacy and reduce the toxicity of this antibiotic [232]. The <5 μm microparticles cross-linked with terephtaloyle chloride (4.5% *w*/*v*) were synthesised and found to encapsulate amikacin. Zhao et al. [226] synthesised a new HP-β-CD polymer (with the addition of epichlorohydrin), which exhibited approximately 46.6 times higher water solubility than pure HP-β-CD. Next, ICs of **azithromycin** with HP-β-CD and its polymer were prepared, which led to a 9- or 18-fold increase in solubility, respectively, compared to sole azithromycin.

Polymer microspheres of β-CD were obtained by inverse emulsion polymerisation with epichlorohydrin as a cross-linking agent and loaded with **ornidazole** [251]. The release of the drug from the synthesised microspheres, which lasted 9 h, showed good sustained, but pH-dependent behaviour. The cumulative release rate of the microspheres was the largest at the pH value of 6.8 (91%) and the smallest in the acidic environment at pH 1.2 (33%). The release profile showed the best correlation with the Ritger–Peppas model, indicating that they are consistent with Fickian diffusion at low pH and non-Fickian diffusion at high pH (a combination of drug diffusion and skeletal corrosion). In addition, the results of the antibacterial test against *E. coli* and *S. aureus* confirmed the maintenance of the antibacterial activity of ornidazole.

Another drug-delivery system was based on nanospheres made of hexanoyl-β-CD and **metronidazole** by adding an acetone amphiphilic CD solution to an aqueous solution of metronidazole with or without Pluronic PE/F68^®®®^ as the surfactant [248]. An optimised formulation for intravenous administration with high encapsulation efficiencies and appropriate particle size (<1 µm) was developed.

A new approach of **cefadroxil**/β-CD-based nanosponges (NSs) with diphenylcarbonate for the cross-linking was developed to prevent hydrolysis of the lactone ring and prolong the drug release to achieve the desired serum level [65]. XRPD, DSC, and FT-IR methods confirmed the interactions of cefadroxil with the NS, while in vitro studies showed a sustained release over 24 h.

**Norfloxacin**-loaded NSs were developed based on β-CD, using diphenyl carbonate as a cross-linking agent to improve the physicochemical properties of the drug [185]. A 1:2 CD:cross-linker proportion was the most optimal due to its higher encapsulation efficiency (80%). NS loaded with norfloxacin showed a controlled release of norfloxacin, almost 100% within 150 min in simulated intestinal fluid (pH 6.5). These NSs also showed a greater passage of norfloxacin compared to norfloxacin alone by using the chamber method in both directions, i.e., from mucosa to serosa as well as from serosa to mucosa. The NS formulation also revealed mucoadhesive properties that can enhance norfloxacin absorption, thus improving its antimicrobial activity tested in the caecal ligation and puncture model, which is the most accepted pattern used for experimental sepsis. Rats treated with norfloxacin-loaded NSs presented a smaller number of colony-forming units (CFU) when compared to animals treated with norfloxacin alone.

### 8.4. Graphene Derivatives

A supramolecular nanocomposite consisting of **amoxicillin**, β-CD, and chitosan/sodium alginate/graphene oxide (GO) NPs has been prepared [58]. GO was incorporated into the sodium alginate and chitosan network to synthesise the nanocomposite. Chitosan, as a cationic polysaccharide, and sodium alginate, as an anionic polysaccharide, have a strong electrostatic interaction. GO contains many hydrophilic groups that are considered effective in self-assembly through physical interaction with the hydrophilic sodium alginate and chitosan, while β-CD carriers have a hydrophilic group that physically interacts with GO. ICs of amoxicillin/β-CD were obtained by three techniques, with the microwave method being the most suitable for the preparation of ICs for the sustained release of amoxicillin. It was experimentally observed that drug release increases with the increase in pH from acidic (pH 2) to neutral (pH 7.0) and decreases from neutral (pH 7.0) to basic (pH 7.4). The drug-release mechanism was explained by both diffusion and relaxation using the Korsmeyer–Peppas equation for each pH level, whereas the application of the Peppas–Sahlin model suggested that the contribution of diffusion was greater than that of the relaxation process.

### 8.5. Inorganic NPs

Akbar et al. [52] enhanced the efficacy of **ampicillin**, **ceftriaxone**, and other compounds (quercetin, naringin, and amphotericin B) against multidrug-resistant bacteria by encapsulating the drugs into β-CD attached to NPs of zinc oxide (ZnO) to form the ZnO-CD-drug complex. The conjugates showed a decrease of MIC for different bacteria as well as minimal cytotoxicity to human cells.

Computational studies were performed to predict that γ-CD is the preferred oligosaccharide that allows for the strongest interaction with **chloramphenicol** in the IC [126]. Consequently, chloramphenicol was attached to the γ-CD-capped silver NPs, showing the synergistic antibacterial activity against *P. aeruginosa* (the most sensitive to this system), *E. faecalis*, *K. pneumoniae,* and *S. aureus*.

### 8.6. Nanofibres

Two nanofibre (NF) systems for **tetracycline** delivery for the treatment of periodontal disease were prepared using the electrospinning process: (i) polycaprolactone (PCL) loaded with tetracycline and (ii) PCL with tetracycline/β-CD complex [113], and 93% and 100% of the tetracycline was released during 14 days from PCL/tetracycline and PCL/tetracycline/β-CD systems, respectively. The results indicated that the process of diffusion from the PCL matrix is different for tetracycline and its complex. However, NFs with ICs appeared to be better protected and increased the biological absorption of tetracycline. In another approach, **tetracycline**/HP-β-CD ICs were mixed with water-soluble nontoxic biopolymer pullulan and, finally, NF networks were obtained via the electrospinning technique. The study demonstrated increased water solubility and a faster release profile of pullulan/tetracycline/HP-β-CD NFs (~55% in 30 s, with a maximum release of 94% in 2 min, and a steady release profile up to 10 min) compared to the pullulan/tetracycline system (65% maximum release in 8–10 min). The release profile of pullulan/tetracycline/HP-β-CD NFs did not fit zero/first-order and Higuchi models, suggesting that tetracycline is not released in a time-dependent manner from the water-insoluble planar matrix. On the other hand, the relatively higher R^2^ in the Korsmeyer–Peppas model suggested that irregular/non-Fickian diffusion and erosion-controlled release of tetracycline from pullulan/tetracycline/HP-β-CD NFs was observed. In addition, the pullulan/tetracycline/HP-β-CD NFs readily disintegrated when wetted with artificial saliva, while the pullulan/tetracycline NFs were not completely absorbed in the same simulated environment. Pullulan/tetracycline/HP-β-CD NFs showed promising antibacterial activity against *S. aureus* and *E. coli* [108].

ICs of α-CD and β-CD with **ciprofloxacin** at room temperature or under the influence of sonic energy were incorporated into NFs via electrospinning based on PCL. The controlled release of ciprofloxacin from the PCL NFs at pH 7.2 was as follows: β-CD/ciprofloxacin (sonic energy) > α-CD/ciprofloxacin (sonic energy) > β-CD/ciprofloxacin (room temperature) > α-CD/ciprofloxacin (room temperature) > ciprofloxacin [160]. The release of ciprofloxacin from PCL NFs was increased due to the increased solubility of ciprofloxacin through the formation of an IC, but its diffusion was controlled by CD, so the resulting NF showed a controlled release pattern for ciprofloxacin. In another approach, ciprofloxacin/β-CD ICs were prepared by the co-precipitation method and then encapsulated into micro- and nano-fibres of polylactic acid (PLA) via the electrospinning technique [162]. The IC improved the loading efficiency of ciprofloxacin into the polymer, while the drug release in PBS 7.4 was more sustained. The ciprofloxacin/β-CD complex was loaded onto the fibre surface, resulting in an initial breaking release phase in the first 3 h, and a more controlled secondary phase of 5–8 days, when the drug loaded into the fibre mass was released. PLA/ciprofloxacin/β-CD IC was fitted to a first-order model. In such a release process, the first stage is a burst effect due to superficial desorption, and as the release progresses, the slope approaches zero, indicating that less ciprofloxacin is released over time. In another approach, electrospinning was used to prepare fast-dissolving gelatine NFs containing ciprofloxacin/HP-β-CD (1:1) ICs obtained by freeze-drying [161]. Computational modelling (molecular docking and molecular dynamics) showed that van der Waals forces were the most important driving forces for complexation and that the hydrophobic moiety (piperazinyl) of ciprofloxacin was included in the HP-β-CD cavity, whereas the hydrophilic moiety of the drug was facing outward from the cavity. The gelatine NF mat loaded with ciprofloxacin/HP-β-CD ICs exhibited fast dissolution (3 s) compared to the gelatine/ciprofloxacin NF mat, which was not able to demonstrate complete dissolution even after 1440 min. This phenomenon is most likely due to the increase in solubility and wettability provided by CD in the mat.

Another study reported a so-called “click reaction”, which was performed between mono-6-propargylamino-6-deoxy-β-CD and azidated cellulose fibres (CFs) to produce covalent grafting of β-CD with the included ciprofloxacin and CFs [163]. The amount of ciprofloxacin HCl incorporated into the product was increased ~2.5-fold compared to unmodified CFs, the release time was extended, and the antibacterial activity against *E. coli* and *S. aureus* was much higher and more sustained, compared to the unmodified CFs.

An ultrasonic bath was used to prepare a ternary system consisting of α-CD and **ofloxacin** in a 1:1 molar ratio with PEG (20% *w*/*w*) as a compatible solubilising agent, thus enhancing complex formation and water solubility [150]. This result can be explained by the increased number of hydroxyl groups in the reaction medium in the presence of PEG, leading to an increased interaction with ofloxacin. The obtained ternary system was loaded into poly(vinyl alcohol) (PVA) with 10% (*w*/*w*) citric acid by the electrospinning method, to synthesise NFs of PVA/α-CD/ofloxacin/PEG. The release of ofloxacin from the PVA NFs was determined, with the results showing a stable release rate of ofloxacin from the NFs over 23 h. The antibacterial effect of NFs against *E. coli* and *S. aureus* was also investigated.

In a different approach, Kim et al. [216] used the electrospinning process to develop fast disintegrating and dissolving NFs consisting of PVA, HP-β-CD, and D-α-tocopheryl polyethylene glycol succinate (TPGS) for local delivery of **roxithromycin** in respiratory tract infection. The role of HP-β-CD was to increase the solubility of the drug, while the incorporation of TPGS into PVA-based NF was to accelerate wetting, disintegration, the dissolution rate, and overcoming bacterial resistance. Moreover, TPGS may accelerate the dissolving speed of the NF mat and overcome the bacterial resistance by inhibiting efflux pumps. The released profiles of roxithromycin from NFs with and without TPGS in the pH 6.8 buffer mimicking the pH of artificial saliva fluid were 84% and 46%, respectively, at 10 min. The observed drug-release data suggested that more than 80% of the drug amount can be released from NFs with the TPGS mat in the oral cavity, which may lead to the fast absorption and initiation of pharmacological actions. The obtained NF showed a higher antibacterial potential in the disc-diffusion method against *E. coli* and *S. aureus* compared to other NF formulations. The antibacterial potential of the PVA/HP-β-CD/TPGS/roxithromycin NF mat was assessed in an in vivo test in a *Streptococcus*
*pneumoniae*-infected mouse model. The authors concluded that the application of this NF to the oral cavity reduced the pneumonia-related disorders in the lung tissue.

**Gentamicin sulphate** shows poor availability to target cells after oral administration. To overcome this obstacle, polyurethane (PU) fibrous membranes were developed with the addition of 15% *w*/*w* of β-CD, functionalised with nanochitosan (5% *w*/*w*) [231]. The amino and hydroxyl groups of chitosan can interact with mucin and open the junction between the epithelial cells, thereby helping in the easy transport of drug molecules into cells. Such optimised multicomponent material showed the best gentamicin release and antibacterial activity compared to sole PU as well as PU functionalised only with β-CD.

Additionally, the solubility of **metronidazole** in water was significantly improved by preparing NF webs of metronidazole ICs with HP-β-CD via electrospinning [246]. These NF webs showed very fast dissolving, reaching a concentration 5 times higher than that of free metronidazole within the first 30 s after being placed in water or in contact with artificial saliva.

### 8.7. Hydrogels

Several studies have focused on the preparation of the **doxycycline**/HP-β-CD complex for the treatment of corneal neovascularisation (CNV) [106,107]. Tetracyclines inhibit matrix metalloproteases responsible for the degradation and remodelling of the underlying basement membrane and prevent pathogenic tissue destruction in CNV. To effectively increase the efficacy of doxycycline, which is not very stable in aqueous solution, complexation with CD was performed. To improve ocular bioavailability, poloxamers P407 and P188 were used to form hydrogels to increase the solution viscosity. NMR and molecular modelling explained that the phenyl group of doxycycline included into HP-β-CD and the hydrophobic interactions were the driving force behind the complex formation [107]. Moreover, it has been suggested that interactions such as hydrogen bonding and/or van der Waals forces between HP-β-CD and the methyl group of doxycycline, as well as a steric effect induced by the HP-β-CD cavity, inhibit the epimerisation and degradation at this site in a competitive manner, thus reducing the formation of degradation products. Meanwhile, NMR spectra indicated that the chelation of Mg^2+^ cations provided a synergetic protection of the unstable site of doxycycline at the N(CH_3_)_2_ moiety. Then, an artificial neural network was developed to predict the optimal conditions for the preparation of doxycycline/Mg^2+^/HP-β-CD complexes based on computational modelling [108]. The highest inclusion efficiency and stability of the complex were obtained using molar ratios of HP-β-CD/doxycycline and Mg^2+^/doxycycline equal to 4 and 10.8, respectively, as well as an inclusion time of 12 h and temperature of 25 °C.

A new composite hydrogel, namely maleated CD-grafted-silylated montmorillonite, has been developed for the specific delivery of **tetracycline** HCl to the colon [112]. Montmorillonite is a type of natural clay mineral with net negatively charged layers and good swelling properties in the presence of water. Maleated CD is well-soluble in water, with unique swelling properties in weak alkaline conditions, and reacts to stimuli based on changes in pH in the gastrointestinal tract. It was demonstrated that this composite hydrogel could successfully deliver tetracycline HCl to the colon without drug loss in the stomach. The release profile of tetracycline from the drug-loaded hydrogel varies with different pH buffers: it is higher at pH 7.4 (~40% within 24 h) than at pH 2.4 (~10% within 24 h). According to the Peppas model, at pH 7.4 there is a swelling-controlled, non-Fickian-type mechanism, while at pH 2.4 there is a diffusion-controlled, Fickian-type mechanism. Thus, the resulting pH-sensitive hydrogels are useful tools for targeting colon-specific diseases in a stimuli-responsive manner based on pH changes in the gastrointestinal tract.

Blanco-Fernandez et al. [153] prepared hydrogels containing HP-β-CD or M-β-CD complexes with ciprofloxacin cross-linked with ethyleneglycol diglycidylether, with or without the addition of agar. Agar promoted the loading of the zwitterion ciprofloxacin and provided the hydrogels with the ability to retain the drug in an aqueous environment at low pH. The reason for this is the electrostatic interaction between ciprofloxacin and agar in water, even though the network is highly swollen. Release was only triggered by the pH shift to 7.4, leading to a 12 h sustained drug delivery.

Another polymer network was created with the use of sterculia gum, which is a complex, branched, and partially acetylated polysaccharide, consisting of glucuronic acid, galacturonic acid, and carbopol [154]. The presence of CD in hydrogels affects both drug entrapment and the release of drugs from the drug-loaded hydrogels. Ciprofloxacin entrapment was greater in the case of hydrogels with CD (increased 1.15 times) compared to hydrogels without CD; however, the release of ciprofloxacin from β-CD-containing hydrogels was slower compared to non-β-CD hydrogels. In this system, drug release was controlled by one or more of the following processes: solvent transport into the polymer matrix, swelling of the bound polymer, diffusion of the solute through the matrix, and erosion/relaxation of the swollen polymer.

Gauzit Amiel et al. [26] prepared water-soluble poly(cyclodextrin citrate) by polymerisation between β-CD and citric acid in the presence of a catalyst of sodium hypophosphite for a local release of ciprofloxacin in diabetic foot infections. Freeze-dried physical hydrogels were obtained, consisting of two oppositely charged polyelectrolytes, chitosan as a natural cationic polysaccharide and poly(cyclodextrin citrate) as an anionic polysaccharide, forming a network in various proportions (3:0, 3:1, 3:3, 3:5, and 3:7 ratio). The results showed that the release of ciprofloxacin decreased as the poly(cyclodextrin citrate) ratio increased in the sponge formulations due to inclusion in the CD cavities. Thus, the highest amount of ciprofloxacin was observed after 24 h for sponges in a ratio of 3:0 after the thermal treatment. Such sponges with a greater swelling rate absorbed and released the highest doses of drugs with a slow sustained kinetic profile. Moreover, a significant difference in the amount released was observed between sponges of the same ratio with and without the heat treatment. The release was greater when devices were not heat-treated. Kinetic release patterns showed a profile with an initial burst followed by a plateau [148].

Recently, self-assembled supramolecular hydrogels based on polypseudorotaxane systems were prepared for delivery of ciprofloxacin to the eye, mixing a drug-loaded dispersion of poloxamer 407 with α-CD solution under continuous agitation [155]. A significant improvement in the antibacterial activity of ciprofloxacin was shown in the optimised formulation, and more drug was retained in the pre-corneal area of the eye for a longer time compared to a conventional gel. Both in vitro and ex vivo permeation studies have shown increased permeability compared to the Carbopol gel available on the market. The antibacterial studies against *S. aureus*, *E. coli*, and *P. aeruginosa* showed an improvement in the antimicrobial activity of ciprofloxacin in a supramolecular hydrogel [155].

A new formulation was proposed for the temperature-sensitive in situ gel, consisting of 0.5% (*w*/*v*) **azithromycin**, the equimolar amount of HP-β-CD, 18.91% (*w*/*v*) of poloxamer 407, and 0.35% (*w*/*v*) of Carbopol 934P [227]. This formulation, optimised with regard to percentage drug release at the first hour, gelation temperature, and viscosity, showed the sustained drug release over a 54 h period, which may be effective in the treatment of periodontal disease.

Simões et al. [240] obtained supramolecular gel formulations, consisting of **vancomycin** HCl (5.5 mg/mL), α-CD (0%—control, 2.5%, 5.0%, 7.0%, and 9.7% *w*/*v*), and Pluronic F127 (6.5%, 13%, or 20% *w*/*v*). Vancomycin HCl was incorporated to the formulations before Pluronic F127 and α-CD solutions were mixed. This drug was physically entrapped in the supramolecular gel and did not interfere in the Pluronic F127–α-CD interaction. Gel formulation without α-CD released 80% of the drug during the first 24 h, while that prepared with 9.7% α-CD released less than 45% in the same time period. In this last case, 100% release was achieved at day 7. Meanwhile, the Pluronic F127 concentration did not significantly affect the vancomycin release rate. The gels prepared with 6.5% copolymer and 5% or more α-CD were stable for at least two months when stored at 20 °C and showed activity against *S. aureus*.

New drug-delivery hydrogels based on the β-CD polymer with the addition of cross-linkers such as 2-isocyanatoethyl 2,6-diisocyanatohexanoate or diisocyanatohexane have been developed [237]. The gels were loaded with rifampicin, **novobiocin**, and **vancomycin**. The release of vancomycin from these gels was slower than from dextrose gels, as dextrose is chemically similar to CD but incapable of forming a complex with a drug, leading the authors to the conclusion that CD ICs were formed. The release was faster with novobiocin and vancomycin than with rifampicin due to their higher solubility in release medium [237].

### 8.8. Other Nanosystems

**Cefradine** was combined with a medusa-like β-CD with 1-methyl-2-(2′-carboxyethyl) maleic anhydrides to improve drug delivery to specific targets, including weakly acidic tissues or organelles, such as tumours, inflammatory tissues, abscesses, or endosomes [66]. Interestingly, amide derivatives of maleic acid were used for conjugation with a drug targeting weakly acid environments, due to their rapid degradation at an appropriate pH, leading to the so-called pH-sensitive drug release. At pH 7.4, less than 20% of cefradine was released within 5 h. On the other hand, a rapid burst was observed at pH 5.5, where more than 80% of cefradine was released within 0.5 h. Another study characterised a sustained-delivery system for periodontitis, consisting of a **doxycycline**/α-CD complex loaded into a PLGA-based implant injectable in situ to reduce irritation and improve the stability of doxycycline in the aqueous medium [109]. The results show that the formulation provides sustained release (96% over 14 days) with a minimum initial burst. The data fit well with the Higuchi model, indicating that the main mechanism follows diffusion.

In another approach, a γ-CD metal-organic framework (MOF) loaded with **enrofloxacin** and **FF** was prepared [131]. The γ-CD-MOF was prepared via an ultrasonic method to obtain small particles. Next, due to the porous structure of γ-CD-MOF, FF and enrofloxacin were absorbed into the pores of γ-CD-MOF, depending on the antibiotics/γ-CD-MOF ratio. The system showed better activity against bacteria than free antibiotics (lower MIC value against *S. aureus* and *E. coli*), released enrofloxacin, and FF over a significantly extended period of time (80% FF, and 87% enrofloxacin over 4 h). The results indicated that the porous structure of γ-CD-MOF controls the release behaviour of both drugs. In addition, antibiotic-loaded γ-CD-MOF showed better long-lasting activity (longer antibacterial time), nontoxicity, as well as excellent biocompatibility with mammalian cells and tissues both in vitro and in vivo.

Next, a β-CD covalent organic framework (COF) loaded with enrofloxacin was produced, obtained by blending the drug and COF into a solution, and incorporated into thermoplastic polyurethane fibres via electrospinning [189]. The drug release involved two typical phases: the enrofloxacin was rapidly released from the membranes within the first 3 h; subsequently, the release dropped sharply and achieved zero-order release. The authors presented the physicochemical and biological properties of the fibrous membrane such as the morphology, hydrophobicity, water uptake capacity, and biocompatibility. The membrane displayed antibacterial activity against *S. aureus* and *E. coli* with 99% inhibitory efficiency for 50 h.

Polyurethane/β-CD/**ciprofloxacin** composite films were developed for use in medical coatings with antibacterial properties [158]. The tested ciprofloxacin/polyurethane films showed antibacterial activity against *S. aureus*, *Sarcina lutea*, *B. cereus*, and *B. subtilis*, as well as Gram-negative bacteria *E. coli* and *P. aeruginosa*. Novel ternary polyurethanes simultaneously containing β-CD and β-glycerophosphate groups (calcium salt, Ca-Glic, or H-Glic) cross-linked by HDI have been designed, using the properties of both polyalcohols: the inclusion capacity of β-CD toward hydrophobic drug molecules and the bioactivity of β-glycerophosphate [159]. The results showed that the CD cavities remained active toward the inclusion of guest molecules. Two polymers were evaluated, one prepared from H-Glic and the other from Ca-Glic. In both cases, there was an initial slow stage (0–5 h) followed by an increase in release up to 10 h, an intermediate plateau between 10 and 30 h, another increase, and a final plateau up to 80 h where 90% of the drug was released. The release profiles were similar, with one exception that the final% release for the Ca-Glic polymer was higher compared to the H-Glic, while the free drug at the same time reached 97%.

In another approach, new macromolecules were synthesised in a reaction between ciprofloxacin and β- or γ-CD, using chloroacetyl chloride as a linking agent [164]. The designed systems showed a pH-dependent prolonged release time and almost five times higher bactericidal activity than the standard ciprofloxacin. The drug-release results showed that the release of ciprofloxacin was maximal and controlled at pH 1.5 (98%), while at pH 7 it was only 20% in 14 h and about 50–60% at pH 4–6.

Garcia-Fernandez et al. [166] developed polyester vascular grafts modified with polymerised M-β-CD and loaded with ciprofloxacin to prevent postoperative infections. The release of ciprofloxacin from the prostheses without poly-M-β-CD was faster than from the M-β-CD functionalised prostheses, regardless of the flow rate, indicating the superiority of poly-M-β-CD in the controlled, sustained release of ciprofloxacin.

A modern delivery system for ciprofloxacin was developed in combination with the HP-γ-CD polymer with the addition of citric acid as a cross-linking agent, grafted onto a woven polyethyleneterephtalate (PET) vascular prosthesis [167]. The CD coating of vascular prostheses may be suitable for the controlled release of ciprofloxacin, which avoids an excessively short and insignificant period of protection from the overly rapid desorption from the graft. The strategy involved grafting HP-γ-CD with ciprofloxacin onto vascular grafts, basing on the pad-dry-cure textile finishing method [167].

Hunag and Kang [195] synthesised polyamidoamine (PAMAM) dendrimers of generations G2 and G4 conjugated with β-CD as a delivery system for **levofloxacin** lactate. PAMAM dendrimers have well-defined structures that allow for precise control of the size, shape, and terminal group functionality, enabling potential pharmaceutical applications such as a drug-delivery vessel. This work is mainly concerned with understanding the host–guest supramolecular behaviour using NMR and IR spectroscopy. Another strategy is the development of polypropylene mesh devices, grafted in two steps with HDI and β-CD, and then loaded with levofloxacin HCl [196]. The antibiotic-loaded mesh samples demonstrated sustained antibacterial properties for 7 and 10 days against Gram-negative and Gram-positive bacteria, respectively. The β-CD-captured levofloxacin HCl showed a burst release after 6 h, a stable release after 9 h, and finally, a sustained release of 84% for the next 48 h.

In another study, Thatiparti et al. [238] described the formation of pseudopolyrotaxane, generated under mild (aqueous, room temperature) conditions, in the synthesis of β-CD polymer after adding diglycidyl ether cross-linkers consisting of ten ethylene glycol units. The formation of these novel affinity polymers resulted in improved mechanical properties, and the affinity-based delivery of **vancomycin** was significantly better than that of the diffusion-only control polymers. The release of drug from dextran and CD-based polymers in PBS at pH 7.4 showed rapid, burst release for vancomycin-loaded dextran, while from CD polymers showed a reduced burst followed by a longer sustained release period, varying with the length of the cross-linker. The bactericidal activity of vancomycin-loaded CD polymers against *S. aureus* continuously showed a zone of inhibition up to 20 days, while the dextran polymers were only able to clear bacteria for 10 days. Next, a vancomycin-delivery nanosystem was developed based on supramolecular amphiphiles that can self-assemble into nanostructures [239]. A novel sugar-based cationic amphiphile derivative with a β-CD head and long C18 carbon chain with a terminal amine (oleylamine, OLA) was synthesised and formulated using a simple suspension method. The hydrophobic tail of OLA was entrapped in β-CD via inclusion complexation, and the release of vancomycin from these nanovesicles was found to be 80% within 48 h. The slower release of vancomycin from vesicles compared to the drug alone was attributed to the drug entrapped in the core of the vesicles, leading to the drug partition through the vesicle matrix prior to release. The Weibull model was found to be the best fit model, indicating that the release mechanism of vancomycin from nanovesicles was Fickian diffusion. Moreover, the MICs against *S. aureus* and MRSA were 2- and 4-fold lower, respectively, compared to vancomycin alone.

### 8.9. Formulations

**Ceftiofur** acid was described as a long-acting injection in patent CN103230364A [94]. The formulation, with ceftiofur acid and HP-β-CD in a molar ratio of 1:1, sodium alginate, and poloxamer 407 and 188, allowed for a 72.7-fold increase in solubility (from 0.03 to 2.18 mg/mL).

Xu et al. [127] formulated new eye drops based on the ICs of **chloramphenicol** with SBE-β-CD, characterised with higher topical concentrations, due to increased solubility and reduced penetration through the cornea (demonstrated using a Franz diffusion cell system). The tear fluid elimination kinetics study showed that the chloramphenicol/SBE-β-CD eye drops significantly extended the residence time, increased the concentration of antibiotic in the conjunctival sac, and did not irritate the eyes of rabbits.

Bozkir et al. [156] developed an eye drop formula with **ciprofloxacin** and HP-β-CD, in which the drug solubility tripled at pH 5.5 and doubled at pH 7.4. The complex was more stable at pH 5.5 than at pH 7.4, and the stability constant of the complex was increased by the addition of 0.1% (*w*/*v*) polymer (HPMC and/or PVP) to the aqueous medium. In a further study, ophthalmic in situ gelling systems containing ICs of ciprofloxacin hydrochloride with HP-β-CD at a 1:1 molar ratio were developed to reduce pre-corneal elimination and improve the bioavailability and therapeutic response [157]. As a result, the developed formulations had increased viscosity, exhibited gelation at eye temperature, proved to be therapeutically efficient, and provided sustained drug release over a period of 8 h.

**Rufloxacin** was used in combination with CDs and HPMC in the formulation of ophthalmic solutions [170]. The addition of 0.25% (*w*/*v*) HPMC to solutions containing rufloxacin/HP-β-CD complexes increased the solubilising effect of this CD, thus reducing the amount of CD necessary for solubilisation of 0.3% (*w*/*v*) rufloxacin. Preliminary pharmacokinetic data in rabbits indicated that the ocular bioavailability of 0.3% (*w*/*v*) rufloxacin solubilised by HP-β-CD was higher than the 0.3% (*w*/*v*) rufloxacin suspension used as a reference.

In other studies, tablet formulations consisting of **norfoxacin**/β-CD complexes and other excipients were developed and assessed before and after compression [186]. The study revealed that the formulation prepared by direct compression with microcrystalline cellulose, stearic acid, magnesium stearate, and Aerosil^®®®^ showed the highest dissolution rate when using crospovidone, achieving 98% drug release.

Interestingly, HP-β-CD overcomes the incompatibility of the combination of **ceftazidime** and **vancomycin** in antibacterial formulation for the treatment of bacterial keratitis [77]. The formulation inhibited the incompatibility of both antibiotics at pH greater than 7.3 and both antibiotics maintained unchanged antibacterial activity.

## 9. Conclusions

This article provided an overview with a discussion of 212 literature reports on the delivery of antibiotics and antimicrobial agents’ inclusion in CDs. One third (70) of the cited articles discussed the impact of CD complexation on antibacterial activity; however, there are relatively few studies (10) on increasing the permeability through biological membranes.

In the case of beta-lactams, the main goal of CD complexation was to improve the chemical stability (in 23 out of 61 papers), since the strained beta-lactam ring is prone to degradation reactions such as amide bond hydrolysis. The problem of chemical instability also concerns the tetracycline group (in 6 out of 14 reports), where the tertiary amine and methyl group are the labile fragments of the molecules, and application of CDs seems to be a promising remedy. The main reason for the preparation of chloramphenicol/CD complexes was the low water solubility (in 7 out of 12 articles) and the desire to reduce the toxicity of chloramphenicol. Another clinically significant advantage of the doxycycline/CD complexes is the increased effectiveness in combating bacterial biofilms. Meanwhile, the quinolones are the most studied group regarding CD complexation, with a total of 74 scientific papers mostly on the preparation of multi-component delivery systems. On the other hand, the inclusion of macrolides (16 reports), aminoglycosides (3 reports), and glycopeptides (9 reports), which are compounds with large molecules, into the CD cavity is less likely. However, contradictory statements appear in various articles, i.e., some authors state that inclusion does takes place and others that it does not. Therefore, further research is needed, especially in the field of antibacterial macromolecules, to determine the possible geometry of the combinations and to pinpoint the interactions that hold the drug and CD together.

To obtain a controlled and sustained release of the loaded antibiotic and prolonged systemic circulation, more sophisticated drug-delivery systems were developed, including nanoparticles, microparticles, liposomes, nanosponges, nanofibres, hydrogels, and macromolecules.

Sustained release and targeted release are important directions for the development of a next generation of personalised therapies, providing powerful means of safe antibiotic therapy, offering an opportunity to reduce antibiotic resistance. Targeted drug release has grown in importance as medicine moves towards precision therapy. Such a delivery system crosses biological barriers to reach the target organ, tissue, or cell prior to drug release. In this way, the bioavailability is improved, and the damage caused by binding to the inappropriate sites is reduced. An example can be both doxycycline and tetracycline directed against colon diseases, which were designed in such a way that the release of the drug from the nanosystem takes place at a certain pH value in the gastrointestinal tract, which is an unquestionable advantage for safe pharmacotherapy.

Thus, the interest in CDs is growing steadily and this trend will continue in the coming years due to their versatility and fascinating ability to improve pharmaceutical formulations in the design of innovative therapies. This article differs from many other reviews in that it describes the current state-of-the-art from the perspective of individual drugs, thus showing where much research has been carried out so far and pointing to new lines of research in the field of antibacterial drugs.

## Data Availability

Not applicable.

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
