# Peer review of "Cyclodextrin Inclusion Complexes with Antibiotics and Antibacterial Agents as Drug-Delivery Systems—A Pharmaceutical Perspective"

_pharmaceutics, 2022, doi:10.3390/pharmaceutics14071389_

Round 1

Reviewer 1 Report

In this review article “Cyclodextrin inclusion complexes with antibiotics and anti-bacterial agents as drug delivery systems – a pharmaceutical perspective"), Boczar and Michalska cover developments in the use of cyclodextrins to improve the pharmokinetic and biological properties of antibiotics. In contrast to previous reviews, this review is quite comprehensive.

This paper, which is generally well written, should be of interest those interested in antimicrobial discovery, pharmacology, and glycobiology. However, there are several that if addressed would greatly improve the article. Many of these relate to the length and readability. The article also lacks conclusions and future directions that would expand the review beyond an indexing of available literature.

Concerns

1. This paper is extremely long. There should be an effort made to make the review more concise and focus on the most germane information.

2. The introduction continues for several pages before introducing the specific topic of the paper. It would be helpful to move this early and move more of the information about the uses of CD to a separate section.

3.  Individual papers are described well in the paper, the conclusions that can be drawn from the work on the individual antibiotic classes and the future work that needs to be done or could lead in a promising direction is not clear. This is through throughout the paper, in the conclusion section, and in the abstract. Take-away messages and conclusions as well as open questions and future directions need to be added to this paper.

4. There are quite a few page-long paragraphs throughout. These should be broken up to improve readability.

Author Response

Manuscript Number: pharmaceutics-1768075 

Type of manuscript: Review

Title: Cyclodextrin inclusion complexes with antibiotics and antibacterial agents as drug delivery systems – a pharmaceutical perspective Authors: Dariusz Boczar, Katarzyna Michalska * Dear Editor of Pharmaceutics, Dear Reviewers, We would like to kindly thank You for your and Reviewers’ thorough review that helped us to improve our paper. We have taken into account all suggestions and have made the necessary changes. Our responses are as follows:

Reviewer 1

Comments and Suggestions for Authors

In this review article “Cyclodextrin inclusion complexes with antibiotics and anti-bacterial agents as drug delivery systems – a pharmaceutical perspective"), Boczar and Michalska cover developments in the use of cyclodextrins to improve the pharmokinetic and biological properties of antibiotics. In contrast to previous reviews, this review is quite comprehensive.

This paper, which is generally well written, should be of interest those interested in antimicrobial discovery, pharmacology, and glycobiology. However, there are several that if addressed would greatly improve the article. Many of these relate to the length and readability. The article also lacks conclusions and future directions that would expand the review beyond an indexing of available literature.

Thank you very much for your suggestions.

Concerns

  1. This paper is extremely long. There should be an effort made to make the review more concise and focus on the most germane information.

Response: Thanks a lot for the suggestion. The following changes have been made to make the article shorter:

- 10 schemes of individual drugs were replaced by one scheme, summarizing general formulas for the classes of antibiotics and antibacterial agents;

- some information given simultaneously in the tables and in the main text have been removed from the main text (i.e. which techniques were used to characterize the complex);

- in the cases, when several drugs was mentioned in one article, the information appears in the text only once and is not repeated in other sections;

- long drug names have been replaced by abbreviations;

- similar reports were grouped when it was possible without the loss of information;

To improve the readability of the text, advanced drug delivery systems were moved to a separate section (8. Modern drug delivery system), as requested by Reviewer 3, and grouped accordingly. In addition, the introductory part has been divided into four sections, in which, in a logical order, starting with outlining the purpose of using CD and their derivatives as drug carriers (1.1), presenting the multidirectional possibilities of CD use (1.2) and expectations for drug delivery systems containing antibiotics and antibacterial agents (1.3) are presented.

However, in accordance with the suggestions of Reviewer 2, the kinetics of the release process along with the indication of the drug release mechanisms had to be added in Chapter 8. As a consequence, the reductions made in the text were practically abolished due to the necessity to introduce new information. Ultimately, the manuscript was reduced from 77 (including 66 text pages, 11 pages references) to 70 (56 text/70) pages.

  1. The introduction continues for several pages before introducing the specific topic of the paper. It would be helpful to move this early and move more of the information about the uses of CD to a separate section.

Response: According both to Your and 2nd Reviewer’s suggestions, the introduction was divided into several subsections: 1.1. The use of CDs and their derivatives as drug carriers, 1.2. New possibilities of using CDs and 1.3. Expectations for the drug delivery systems containing antibiotics and antibacterial agents.

  1.  Individual papers are described well in the paper, the conclusions that can be drawn from the work on the individual antibiotic classes and the future work that needs to be done or could lead in a promising direction is not clear. This is through throughout the paper, in the conclusion section, and in the abstract. Take-away messages and conclusions as well as open questions and future directions need to be added to this paper.

Response: The conclusions were changed to indicate the direction of changes in the future.

I present the last amended part of the conclusions: “Sustained release and targeted release are important directions for the development of a next generation of personalized therapies, providing powerful means of safe antibiotic therapy, offering an opportunity to reduce antibiotic resistance. Targeted drug release has grown in importance as medicine moves towards precision therapy. Such a delivery system crosses biological barriers to reach the target organ, tissue, or cell prior to drug release. In this way, the bioavailability is improved and the damage caused by binding to the inappropriate sites is reduced. An example can be both doxycycline and tetracycline directed against colon diseases, which were designed in such a way that the release of the drug from the nanosystem takes place at a certain pH value in the gastrointestinal tract, which is an unquestionable advantage for safe pharmacotherapy.

Thus, the interest in CDs is growing steadily and this trend will continue in the coming years due to their versatility and fascinating ability to improve pharmaceutical formulations in the design of innovative therapies”.

  1. There are quite a few page-long paragraphs throughout. These should be broken up to improve readability.

Response: The page-long paragraphs appeared mainly in the discussion of chloramphenicol. They have been divided into smaller ones.

Reviewer 2 Report

Reviewer comments: To date, there are many excellent review articles available on native CDs and their derivatives, their well-known structural features, physicochemical properties and possible applications”. The authors can summarize each of them. 

Reviewer comments: What is the difference between what has been published with what the authors want to publish? 

Reviewer comments: Introduction section is too long. The authors must do an introduction shorter. 

Reviewer comments: The authors must describe the main physicochemical and biological properties that should have the drug delivery systems.  

Reviewer comments: What differences and similitudes exist between the cyclodextrin inclusion complexes and other systems such as hydrogels stimuli sensitive, nanoemulsions, liposomes, etc etc?? 

Reviewer comments: (line 226, page 5).Thus, complexes of beta-lactam antibiotics with CDs are desirable to allow the safe use of the sustained-release pharmaceutical form in a single daily dose”. In this sense, why cyclodextrin inclusion complexes can have this function?? What is its mechanics?? How affect pH, temperature, etc etc? . Explain more in respect. 

Reviewer Comments: In each section, the authors describe research related to using cyclodextrin inclusion complexes for antibiotics. However, the description of each author is very poor. In this sense, authors must explain the results of more importance of each research. 

Reviewer Comments: Some advanced drug delivery systems for antibiotics and antibacterial agents would also be good to mentioned, especially in the conclusions section, which is now poorly written. 

Reviewer Comments: At first glance, the article looks like a review on antibiotics since the number of figures related to the molecular structure of this type of drug is abundant, i.e., 10 in total. Figures must be those obtained from the reviewed articles where some of the most important properties are demonstrated, such as drug delivery systems.  

Reviewer Comments. The authors must improve the Table content since they do not provide enough information on the articles described. For example, the authors did not describe the results more relevant. Also, the stability constant and solubility improvement is information irrelevant.  

Reviewer Comments. The authors must group and not describe each article separately; for this reason, the tables are too large. 

Author Response

Manuscript Number: pharmaceutics-1768075 

Type of manuscript: Review

Title: Cyclodextrin inclusion complexes with antibiotics and antibacterial agents as drug delivery systems – a pharmaceutical perspective Authors: Dariusz Boczar, Katarzyna Michalska * Dear Editor of Pharmaceutics, Dear Reviewers, We would like to kindly thank You for your and Reviewers’ thorough review that helped us to improve our paper. We have taken into account all suggestions and have made the necessary changes. Our responses are as follows:

Reviewer 2

Comments and Suggestions for Authors

Thank you very much for your suggestions.

Reviewer comments: “To date, there are many excellent review articles available on native CDs and their derivatives, their well-known structural features, physicochemical properties and possible applications”. The authors can summarize each of them. 

Response: There are a huge number of review articles summarizing the possible structure, types, uses and physicochemical properties of cyclodextrins, and the collection grows every month. In order to shorten the introduction and the entire article, we had to omit the discussion of these works. These works have only been signaled. People interested in the topic were redirected to the most interesting articles on this topic in our opinion.

Reviewer comments: What is the difference between what has been published with what the authors want to publish?

Response: The aim of the article is to present a possibly complete review of literature reports on the complexation of antibiotics and antibacterial drugs with CD, and to discuss 212 original papers. This enables a holistic view of the entire group, but also of individual representatives of a given class. Our proposed table clearly shows what has already been done for a specific drug. In this way, it avoids breaking down a door that has already been opened by other research groups.

Reviewer comments: Introduction section is too long. The authors must do an introduction shorter.

Response: Thank you very much for your suggestions.

According both to Your and 1st Reviewer’s suggestions, the introduction was divided into several subsections: 1.1. The use of CDs and their derivatives as drug carriers, 1.2. New possibilities of using CDs and 1.3. Expectations for the drug delivery systems containing antibiotics and antibacterial agents.

Reviewer comments: The authors must describe the main physicochemical and biological properties that should have the drug delivery systems.

Response: The main physicochemical and biological properties are summarized in section 1.3. Expectations for the drug delivery systems containing antibiotics and anti-bacterial agents.

Reviewer comments: What differences and similitudes exist between the cyclodextrin inclusion complexes and other systems such as hydrogels stimuli sensitive, nanoemulsions, liposomes, etc etc??

Response: The advanced drug delivery systems described in this paper, such as nanoparticles, hydrogels etc., must contain a complex of antibacterial agent and CD as a main component. The main difference is the drug release profile – an immediate release for simple drug/CD complexes and a sustained release for the more advanced delivery systems.

Reviewer comments: (line 226, page 5). “Thus, complexes of beta-lactam antibiotics with CDs are desirable to allow the safe use of the sustained-release pharmaceutical form in a single daily dose”. In this sense, why cyclodextrin inclusion complexes can have this function?? What is its mechanics?? How affect pH, temperature, etc etc? . Explain more in respect. 

Response: Thank you for this valid suggestion. We supplemented the missing information in the text. Moreover, the sustained release drug delivery systems have been moved to a new, separate section 8. A discussion of the drug release mechanism has been added to the description of the dissolution profiles wherever described in the original work.

Reviewer Comments: In each section, the authors describe research related to using cyclodextrin inclusion complexes for antibiotics. However, the description of each author is very poor. In this sense, authors must explain the results of more importance of each research. 

Response: Thank you for this valid suggestion. We supplemented the missing information in the text. We did our best to describe the most important results of each article from a pharmaceutical perspective: improving solubility, modifying the drug release profile, slowing drug degradation, improving biological membrane permeability, and increasing antimicrobial activity. Additionally, the tables summarize physicochemical information. However, as suggested by the Reviewer, information on the kinetics of the process and the mechanism of this process were also added. Thus, the work has definitely been strengthened. Thanks again for the hint in this regard.

Reviewer Comments: Some advanced drug delivery systems for antibiotics and antibacterial agents would also be good to mentioned, especially in the conclusions section, which is now poorly written.

Response: The separate section has been created for the advanced drug delivery systems and additional discussion of these systems has been added to the conclusion.

Reviewer Comments: At first glance, the article looks like a review on antibiotics since the number of figures related to the molecular structure of this type of drug is abundant, i.e., 10 in total. Figures must be those obtained from the reviewed articles where some of the most important properties are demonstrated, such as drug delivery systems.   

Response: Ten schemes presenting the structural formulas for individual compounds discussed in the paper have been replaced by one scheme, showing only general structure for each class.

Reviewer Comments. The authors must improve the Table content since they do not provide enough information on the articles described. For example, the authors did not describe the results more relevant. Also, the stability constant and solubility improvement is information irrelevant.

Response: The stability constant is a very important quantity, showing how strong are the interactions between the host and the guest. It has an influence on another properties such as complexation efficiency, solubility, dissolution rate, stabilization of a drug molecule, permeability and antibacterial activity.

Solubility is also a very important property. In many cases, solubility is the main factor influencing the dissolution rate and consequently the bioavailability. Due to the low solubility of some antibiotics, their doses have to be high enough to ensure the minimum inhibitory concentration in the body fluids.

The other important properties, not shown in the table, are discussed in the text. Adding them to the tables as additional columns would make them extremely large and would result in many empty cells in the cases when the desired information is not given in the article.

Reviewer Comments. The authors must group and not describe each article separately; for this reason, the tables are too large. 

Response: The tables discuss the physicochemical aspects, whereas the main text focuses on the pharmaceutical perspective.

From the authors’ point of view, discussing each individual paper in a separate row of a columns would make it easier for the reader to make a quick overview of the current state of the art due to the following advantages:

- some experimental details (stoichiometry, characterization techniques, stability constants) are removed from the main text and presented only in the tables;

- the tables gather in one place all the reports concerning the same API throughout the text, including the previous mentions (when one article cited earlier reported the complexation of several different drugs) and those in modern drug delivery systems in section 8;

- it is clearly visible which APIs were studied intensively and which require further investigation;

- the reader can easily distinguish between the papers in which much research has been done (e.g. several different cyclodextrins, preparation methods, characterization techniques) and those of a more limited content;

- based on the tables, the reader can easily extract the references to the reports containing the desired information (e.g. articles in which methylated cyclodextrins were used, those in which cytotoxicity study was performed, reporting the stability constants etc.).

Reviewer 3 Report

This review is well written and organized, however, needs some minor changes to be done before its consideration for publication.

Comments

1. In the abstract, the reviewer suggests reconsidering the last portion as there is no separate discussion of the role of the ternary system with the auxiliary substance and delivery system for the cyclodextrin complex (such as nanosponges, nanofibres, nanoparticles, and liposomes, etc) in the manuscript. Moreover, please include different sections for these instead of discussing them under the same headings.   

2. Various sections in the manuscript do not have any linked references (for instance introductive portion of beta-lactam antibiotics). The reviewer suggests to add the proper references for the ease of the reader's point of view.

3. Author provides structural formulas of different antibiotics in schemes discussed in the text; Reviewer suggests instead of providing structural formulas of all the compounds it’s better to provide figures related to the content summarizing the complete rationale of the manuscript or characterization techniques of developed cyclodextrin complexes. So that manuscript becomes informative from figures perspective and it would be beneficial to the reader.

4. Reviewer suggests to incorporate a separate section for the cyclodextrin inclusion complexes based antibiotics and antibacterial agents that are clinically approved marketed products and future perspectives.

Author Response

Manuscript Number: pharmaceutics-1768075 

Type of manuscript: Review

Title: Cyclodextrin inclusion complexes with antibiotics and antibacterial agents as drug delivery systems – a pharmaceutical perspective Authors: Dariusz Boczar, Katarzyna Michalska * 

Dear Editor of Pharmaceutics, Dear Reviewers, 

We would like to kindly thank You for your and Reviewers’ thorough review that helped us to improve our paper. We have taken into account all suggestions and have made the necessary changes.

Our responses are as follows:

Reviewer 3

Comments and Suggestions for Authors

This review is well written and organized, however, needs some minor changes to be done before its consideration for publication.

Thank you very much for your suggestions.

Comments

  1. In the abstract, the reviewer suggests reconsidering the last portion as there is no separate discussion of the role of the ternary system with the auxiliary substance and delivery system for the cyclodextrin complex (such as nanosponges, nanofibres, nanoparticles, and liposomes, etc) in the manuscript. Moreover, please include different sections for these instead of discussing them under the same headings.

Response: According to your request, a separate section discussing the modern drug delivery systems has been created (section 8).   

  1. Various sections in the manuscript do not have any linked references (for instance introductive portion of beta-lactam antibiotics). The reviewer suggests to add the proper references for the ease of the reader's point of view.

Response: Thank you very much for your suggestions.

The references have been added in the introductory paragraphs of each section.

  1. Author provides structural formulas of different antibiotics in schemes discussed in the text; Reviewer suggests instead of providing structural formulas of all the compounds it’s better to provide figures related to the content summarizing the complete rationale of the manuscript or characterization techniques of developed cyclodextrin complexes. So that manuscript becomes informative from figures perspective and it would be beneficial to the reader.

Response: Ten schemes presenting the structural formulas for individual compounds discussed in the paper have been replaced by one scheme, showing only general structure for each class.

  1. Reviewer suggests to incorporate a separate section for the cyclodextrin inclusion complexes based antibiotics and antibacterial agents that are clinically approved marketed products and future perspectives.

Response: Based on the literature search, there are only a few approved marketed products containing cyclodextrin complexes with antibacterial agents (cefotiam, cefditoren pivoxil, chloramphenicol, and norfloxacin). They have been mentioned in the introduction and properly cited.

Round 2

Reviewer 1 Report

This is a revision of a comprehensive review article (“Cyclodextrin inclusion complexes with antibiotics and anti-bacterial agents as drug delivery systems – a pharmaceutical perspective ") by Boczar and Michalska in which they cover developments in the use of cyclodextrins to improve the pharmokinetic and biological properties of antibiotics. This paper should be of interest those interested in antimicrobial discovery, pharmacology, and glycobiology.

The original version had issues related to the length and readability that limited the impact of the review and was missing conclusions and future directions. The authors have made thoughtful revisions and have addressed my concerns. Their revisions have strengthen the paper.

Reviewer 2 Report

The article can be accepted